# STEVE-EYE: EQUIPPING LLM-BASED EMBODIED AGENTS WITH VISUAL PERCEPTION IN OPEN WORLDS

**Sipeng Zheng[1], Jiazheng Liu[2], Yicheng Feng[2], Zongqing Lu[2,1†]**

[1] Beijing Academy of Artificial Intelligence
[2] School of Computer Science, Peking University
spzheng@baai.ac.cn   fyc813@pku.edu.cn   zongqing.lu@pku.edu.cn

## ABSTRACT

Recent studies have presented compelling evidence that large language models (LLMs) can equip embodied agents with the self-driven capability to interact with the world, which marks an initial step toward versatile robotics. However, these efforts tend to overlook the visual richness of open worlds, rendering the entire interactive process akin to "a blindfolded text-based game." Consequently, LLM-based agents frequently encounter challenges in intuitively comprehending their surroundings and producing responses that are easy to understand. In this paper, we propose Steve-Eye, an end-to-end trained large multimodal model to address this limitation. Steve-Eye integrates the LLM with a visual encoder to process visual-text inputs and generate multimodal feedback. We adopt a semi-automatic strategy to collect an extensive dataset comprising 850K open-world instruction pairs, enabling our model to encompass three essential functions for an agent: multimodal perception, foundational knowledge base, and skill prediction and planning. Lastly, we develop three open-world evaluation benchmarks and carry out experiments from a wide range of perspectives to validate our model's capability to strategically act and plan. The project's website and code can be found at https://sites.google.com/view/steve-eye.

## 1 INTRODUCTION

Developing embodied agents that can adapt to the open world has long been a substantial challenge (Kolve et al., 2017; Savva et al., 2019). Recently, the rapid progress of large language models (LLMs) (OpenAI, 2022; Touvron et al., 2023a) has shown their potential to serve as a general-purpose assistant. Driven by these pre-trained LLMs, recently proposed agents (Yuan et al., 2023; Wang et al., 2023a;b; Zhu et al., 2023) have managed to extract world knowledge and reasoning capabilities from LLMs, allowing them to become self-driven. Thereby these agents are capable of generating executable policies or plans for a wide range of skills and tasks in an open-ended world.

While current attempts to integrate LLMs show promise in developing a generic embodied agent, these efforts primarily translate the entire world into text, which overlooks the multifaceted richness of diverse visual reality

**Figure 1.** (a) LLM-based agent's feedback is uncontrollable due to the uncertainty of input textual prompt, while visual cues can benefit the agent to generate feedbacks; (b) a text-only driven agent often finds it difficult to produce intuitive feedback that humans can easily understand.

---

[†]Corresponding author.

and turns interacting with the environment into something akin to "**a blindfolded text-based game**." Consequently, such text-only agents often face difficulties when it comes to effectively and intuitively representing the world. Imagine a situation where you request your agent to shop for a pair of shoes. Would you prefer to send the agent a picture of the shoes or provide a lengthy description of the shoes to convey their appearance? Undoubtedly, you would opt for the former choice.

In fact, the agent's reliance on text input/output (I/O) imposes significant limitations on its ability to interact with the world. To illustrate this point, we consider Minecraft (Guss et al., 2019; Fan et al., 2022) as an ideal example. Minecraft, being an expansive sandbox game, offers a vast realm for embodied agents to explore, which requires the acquisition of various basic skills (e.g., crafting logs) and the ability to plan and execute diverse tasks. First, as shown in Figure 1 (a), the LLM-based agent produces uncontrollable outputs. The success of the agent's responses hinges heavily on careful prompt engineering (Huang et al., 2022b), ensuring that the LLM comprehends the environment and task objectives. Moreover, a universally applicable prompt that suits every LLM and task is an unattainable goal. Therefore, this prompting process is labor-intensive and contradicts our aim of enabling agents to act in a self-driven manner. Second, when compared to visual feedback, language often encounters difficulties in intuitively conveying specific world concepts (e.g., recipes) to users, as illustrated in Figure 1 (b), thereby unavoidably creating obstacles for robust human-computer/AI interaction (Preece et al., 1994; Fallman, 2003).

Unlike LLMs, humans possess an innate ability to process and generate information through both visual and text channels. This inherent gift significantly enhances our capability to interact with the world. However, the coupling of LLM-based agents with multimodal I/O has been relatively underexplored in an open-ended environment. To fill this gap, we introduce **Steve-Eye** 🖥️, a large multimodal model that enables LLM-based embodied agents to engage with the open world via visual-text interfaces. Steve-Eye excels at producing responses that demonstrate a comprehensive grasp of the environment, common-sense reasoning, and executable skill plans. To achieve this, Steve-Eye is equipped with three indispensable functions: (1) multimodal perception; (2) foundational knowledge base; and (3) skill prediction and planning. In this paper, we choose Minecraft as our validation platform considering its vast sandbox world and the high degree of freedom. More environments can also be considered, e.g., Virtual Home (Puig et al., 2018), AI2THOR (Kolve et al., 2017). Due to the space limit, we discuss the exploration of more generic environments in Appendix E and leave it as our future work. Our contributions can be summarized as follows:

**Open-World Instruction Dataset.** We construct an extensive instruction dataset to train Steve-Eye for the acquisition of three mentioned functions. The instruction data contains not only the agent's per-step status and environmental features but also the essential knowledge for agents to act and plan. However, collecting such a dataset in an open world can be a costly endeavor, especially when aiming to gather fine-grained and diverse labels. As a result, previous studies (Fan et al., 2022) have often relied on readily available unsupervised data (e.g., video-subtitle pairs) for pre-training. In these approaches, the agent's comprehension of its status and environment is implicitly learned through self-supervised techniques, while its foundational knowledge is directly derived from general-purpose LLMs. In contrast, our work involves curating multimodal instructional data specifically designed for open-ended embodied agents, by utilizing ChatGPT (OpenAI, 2022).

**Large Multimodal Model and Training.** Steve-Eye combines a visual encoder which converts visual inputs into a sequence of embeddings, along with a pre-trained LLM which empowers embodied agents to engage in skill or task reasoning in an open world. During the training process, we employ a two-stage strategy similar to Liu et al. (2023). This strategy commences with the alignment of multimodal elements between the visual encoder and the large language model, followed by the instruction tuning through our constructed dataset.

**Open-World Benchmarks.** We carry out extensive experiments to demonstrate that our proposed Steve-Eye outperforms LLM-based agents in open-world setups. Specifically, we develop the following benchmarks to evaluate agent performance from a broad range of perspectives: (1) environmental visual captioning (ENV-VC), which assesses an agent's capacity to perceive and describe its surroundings effectively; (2) foundational knowledge question answering (FK-QA), which evaluates the proficiency in mastering basic knowledge crucial for an agent's decision-making; (3) skill prediction and planning (SPP), which quantifies an agent's capability to act and plan strategically.

## 2 RELATED WORK

### 2.1 OPEN-WORLD EMBODIED AGENTS WITH LLMS

The rapid progress of large language models (Brown et al., 2020; Raffel et al., 2020; Zhang et al., 2022; Chowdhery et al., 2022) has significantly boosted their capacity to encode a wide range of human behaviors within training data (Bommasani et al., 2021). When equipped with narrowly designed prompts, LLM-based agents exhibit the capability to generate executable plans for tasks such as indoor robot manipulation. For instance, SayCan (Ahn et al., 2022) integrates skill affordances with LLMs to yield actionable plans, while Palm-E (Driess et al., 2023) takes a step further by constructing hierarchical agents capable of handling multimodal prompts. This approach has also proven its efficacy in open-world environments (Huang et al., 2022a; Li et al., 2022). In contrast to robot manipulation, agents in the wild require a heightened level of real-time situational awareness and foundational knowledge to execute intricate skill plans across a diverse array of tasks. To simulate human behaviors in such open worlds, Generative Agents (Park et al., 2023) store agents' experiences and retrieve these memories to generate plans in a text-based sandbox game.

In recent years, the 3D sandbox Minecraft has received considerable attention owing to its remarkably flexible game mechanics to serve as a prominent open-world benchmark (e.g., MineRL (Guss et al., 2019) and Minedojo (Fan et al., 2022)). DEPS (Wang et al., 2023b) introduces the descriptor, explainer, and selector for plan generation with the help of LLM. Plan4MC (Yuan et al., 2023) constructs a skill graph and proposes a skill search algorithm to minimize planning errors. Voyager (Wang et al., 2023a) proposes an LLM-powered lifelong learning agent that continually explores the Minecraft world. Similar to (Park et al., 2023), GITM (Zhu et al., 2023) integrates LLMs with text-based memory and knowledge to create generic agents in Minecraft. Among these studies, Voyager (Wang et al., 2023a) and GITM (Zhu et al., 2023) lean entirely on text descriptions of the environment to act and plan, while Plan4MC (Yuan et al., 2023) and DEPS (Wang et al., 2023b) have visual-input skills but still rely on merely text for planning. None of them try to understand the rich visual observation provided natively by Minecraft. In contrast to these works, our work trains a large multimodal model to fill this gap.

### 2.2 LARGE MULTIMODAL MODELS (LMMS)

In comparison to LLMs, large multimodal models (LMMs) (Awadalla et al., 2023) encompass a broad range of information beyond text modality, which can be categorized into two primary streams. The first category (Gupta & Kembhavi, 2023; Huang et al., 2023a; Patil et al., 2023; Surís et al., 2023) involves hinging on ChatGPT (OpenAI, 2022) or GPT-4 (OpenAI, 2023) to generate in-context responses without parameter tuning. However, these approaches heavily rely on the availability of an LLM's API and the quality of the designed prompts. The second category comprises end-to-end pre-trained models. Within this category, models such as Huang et al. (2023b); Peng et al. (2023) are trained entirely from scratch. Conversely, some research explores efficient fine-tuning using pre-trained LLMs by incorporating lightweight modality encoders, such as Qformer (Li et al., 2023) or Perceiver (Alayrac et al., 2022). Recently, Liu et al. (2023) propose to explicitly instruction-tune a LLM using vision-language instruction data.

In this work, we propose Steve-Eye by building upon pre-trained LLMs, aiming to develop an open-world agent powered by a large-scale model with versatile multimodal I/O capabilities.

## 3 METHODOLOGY

In this section, we first provide our instruction-following dataset to develop three key functions for the agent's open-world interaction in Section 3.1. We then propose our large multimodal agent Steve-Eye in Section 3.2, and clarify details of the training procedure in Section 3.3. We adopt Minecraft as our open-ended platform in this paper to collect data and validate the model, anticipating to explore a broader range of environments for Steve-Eye in future studies.

To empower an agent with the self-driven capacity to act and plan in an open world, we posit that the following embodied functions are indispensable: (1) multimodal perception function which offers a detailed description of the agent status and environmental features; (2) foundational knowledge

base which imparts an understanding of how the world works and conveys crucial basic knowledge related to skills and tasks; (3) skill prediction and planning which is responsible for generating skill execution feedback (e.g., success or failure) and crafting high-level skill plans for handling more complex and long-horizon tasks. We develop these functions by building the corresponding instruction dataset to pre-train Steve-Eye as follows.

### 3.1 OPEN-WORLD INSTRUCTION-FOLLOWING DATASET

**Multimodal Perception Instructions.** Human players can perform actions in Minecraft mainly relying on their visual perception, without any prior hints or imposed game judgments. In order to endow Steve-Eye with the same ability, it is required to provide it with comprehensive visual descriptions of the environment. To achieve this, we use Minedojo (Fan et al., 2022) to obtain Minecraft snapshots which contain a wide array of details within the agent's surroundings, including environmental features, the agent's life and food status, inventory items, and equipment, as illustrated in Figure 2. In addition, we leverage MaskCLIP (Zhou et al., 2022) to identify the in-sight objects of these snapshots without supervised annotations. During our data collection process, for each snapshot $\mathcal{I}$ and its corresponding description $\mathcal{X}_C$, we initiate a three-step approach. Firstly, we prompt ChatGPT to curate a list of 40 instructions as shown in Figure 6 in Appendix A. Then we enrich snapshot details as dense caption to describe its content, with the assistance of ChatGPT. Finally, we select an instruction $\mathcal{X}_Q$ randomly from the list and combine it with the snapshot's caption to create a single-round multimodal description pair (e.g., ### Human: $\mathcal{X}_Q$ $\mathcal{I}$ \n ### Embodied Agent: $\mathcal{X}_C$ \n.). By doing so, we collect 200K instructional pairs for multimodal perception learning.

**Foundational Knowledge Instructions.** Embodied agents require a foundation of essential knowledge to facilitate action-taking and skill planning. In Minecraft, such knowledge should contain item recipes, details of item attributes, their associated numerical value, etc. We access this vital information from Minecraft-Wiki (Fandom, 2023), which comprises an extensive collection of over 9,000 HTML pages. To be specific, we first obtain all item icons from Minecraft-Wiki and generate 200K icon inventory images, as illustrated in Figure 3 (a). Each icon image corresponds to a 4-row table with an associated caption adhering to a standardized template: "There is a Minecraft inventory with 4 rows. From left to right, they

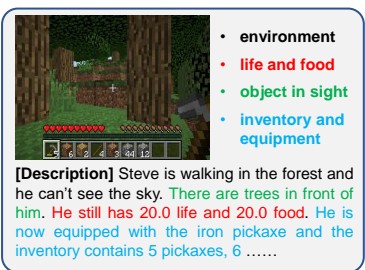

[Description] Steve is walking in the forest and he can't see the sky. There are trees in front of him. He still has 20.0 life and 20.0 food. He is now equipped with the iron pickaxe and the inventory contains 5 pickaxes, 6 ......

**Figure 2.** Multimodal perception

are ...". As shown in Figuire 7 in Appendix A, we curate a set of 20 distinct prompts designed to challenge the model's ability to recognize items. Subsequently, we further collect all recipe-related information from the Wiki as illustrated in Figure 3 (b), and design similar prompt templates to formulate 10,000 recipe-image instructional pairs. Lastly, we process the Wiki and utilize this corpus to produce 40,000 single-round question-answer pairs. In total, we collect a high-quality dataset with 250K foundational knowledge instructions.

**Skill-related Interaction Instructions.** The environmental description and foundational knowledge serve as prerequisites for an agent's interaction within the open world. However, a successful interaction requires more than these elements alone. It relies upon the mastery of basic skills, such as log, harvesting, and food preparation, as well as high-level skill planning abilities to tackle complex, long-horizon tasks, such as crafting an iron pickaxe. To facilitate this, we gather corresponding training data for skill prediction and planning, which enables our model to provide correct feedback on both basic skills and long-horizon tasks across a spectrum of agent or environmental conditions. Specifically, the data collection process involves two steps. First, we sample skill trajectories based on the pre-trained basic skill policies and collect 200K snapshot pairs with corresponding statuses from these trajectories. Each snapshot pair $\{\mathcal{I}_0, \mathcal{I}_t\}$ denotes the 0-th and t-th timestamp of the skill trajectory. Next, we employ ChatGPT to generate question-answer pairs about diverse aspects of skill execution status. These questions

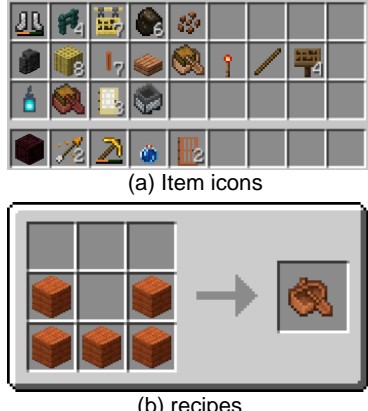

(a) Item icons

(b) recipes

**Figure 3.** Icons and recipes

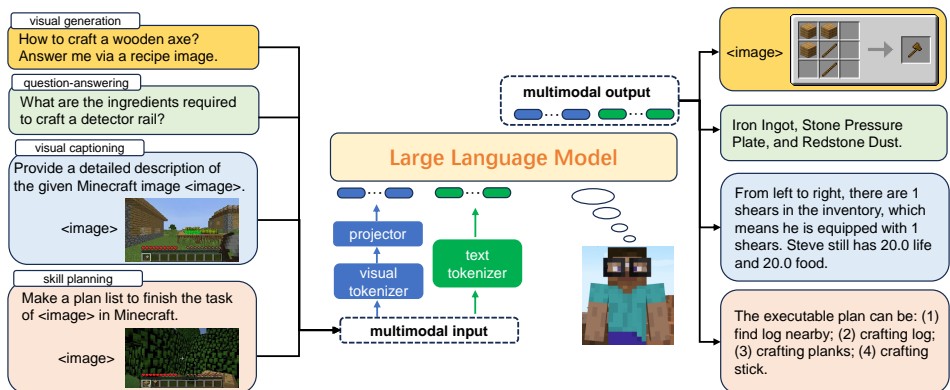

**Figure 4.** Illustration of Steve-Eye: a large multimodal model designed to seamlessly process both visual and language inputs. Steve-Eye excels in acquiring fundamental knowledge of the world it lives in, understanding the nuances of its surroundings, and generating executable plans to complete a wide array of open-ended tasks. Furthermore, Steve-Eye responds to user instructions through either visual or text-based cues, enhancing the convenience and flexibility of human-AI interaction.

delve into whether the agent completes the skill, encounters unexpected failures, or seeks explanations for such failures. More details can be found in Appendix A. Second, we sample 40K task trajectories using the planner in Yuan et al. (2023), each of which can be denoted as $\mathcal{T} = \{s_1, s_2, ...s_T\}$ representing the task is finished via a T-round planning procedure, where $s_i$ is the skill plan for $i$-th round. At each round $i$, we feed our model with its start snapshot and task initialization, and curate instructional questions to inquire about $s_i$ with reasonable explanation. In this manner, we obtain 200K instructional pairs from task trajectories.

## 3.2 MODEL ARCHITECTURE

Figure 4 illustrates the overall architecture of our proposed model. Steve-Eye, functioning as a generative model, connects an image-oriented tokenizer $f_v$ with the pre-trained LLM backbone $\Theta$. We adopt the image tokenizer, e.g., VQ-GAN (Esser et al., 2021), to encode the raw images $\mathcal{I}$ into token embeddings $\mathcal{V} = \{v_1, v_2, ..., v_n\} \in \mathbb{R}^{n \times d}$, where $n$ denotes the number of visual tokens and $d$ is the dimensionality of each token. We further utilize a lightweight projection module $f_l$ with a trainable projection matrix $W$. This module maps the visual tokens to the same space with text embeddings, yielding $\hat{\mathcal{V}} = \{\hat{v}_1, \hat{v}_2, ..., \hat{v}_n\} \in \mathbb{R}^{n \times \hat{d}}$:

$$\hat{\mathcal{V}} = W\mathcal{V}; \text{ where } \mathcal{V} = f_v(I). \tag{1}$$

To effectively process visual-language inputs and generate corresponding outputs, our model integrates the visual codebook $\mathcal{C}_v$ into the pre-existing language vocabulary $\mathcal{C}_l$. This integration leads to the formation of a unified multimodal codebook, denoted as $\mathcal{C}_m = \mathcal{C}_v \cup \mathcal{C}_l$. Additionally, in order to mark the starting and ending points of visual elements in I/O sequences, we introduce two special tokens, namely <vis> and </vis>. The LLM backbone $\Theta$ of our Steve-Eye is built upon a decoder-only architecture with casual transformers. Our model employs an auto-regressive prediction mechanism, generating responses based on the provided multimodal input tokens. The resulting response is a mixed sequence of visual and textual tokens, represented as $\mathcal{Y} = \{y_1, y_2, ..., y_m\}$. For each embedding $y_i$, we pass it through a linear layer $f_p$ followed by a softmax operation, mapping it into a probability distribution of the multimodal vocabulary. The final prediction for the $i$-th token $z_i$ is determined by selecting the token from the multimodal codebook with the highest score:

$$z_i = \arg\max(\text{softmax}(f_p(y_i))). \tag{2}$$

## 3.3 TRAINING

Each instruction-following instance can be formulated as a multi-round conversation $\{\mathcal{X}_Q^1, \mathcal{X}_C^1, ..., \mathcal{X}_Q^N, \mathcal{X}_C^N\}$, where each $\{\mathcal{X}_Q^i, \mathcal{X}_C^i\}$ represents a question-answer interaction between a human and

the embodied agent and $N$ indicates the total number of rounds in the conversation. The entire instructional dataset follows this unified template, as demonstrated in Figure 11 in Appendix A. To efficiently train our model, we employ the negative log-likelihood objective over the prediction tokens with instruction tuning:

$$\mathcal{L}(\Theta) = -\sum_{j=1}^{L} \log P_\Theta(y_j | \mathcal{I}, \hat{y}_{1:j-1}), \tag{3}$$

where $y$ and $\hat{y}$ respectively denote the input and target token sequences, with $\Theta$ representing the model parameters, and $L$ representing the length of the target sequence. The input visual content $\mathcal{I}$ may represent an empty image depending on the input instruction. It is worth noting that we constrain the loss computation to only consider the answer tokens $\mathcal{X}_C$. This constraint prevents training from becoming excessively straightforward and ensures that the model's primary focus is on learning to precisely generate coherent responses. Similar to Liu et al. (2023), we adopt a two-stage instruction-tuning strategy to train our model:

**Two-Stage Instruction-Tuning.** **(1) Multimodal feature alignment**: In the first stage, our primary objective is to align visual features with the language token space. In order to strike a balance between efficient tuning and a comprehensive coverage of the world's concepts, we curate our open-ended instruction dataset to 600K snapshot-text pairs. These pairs are then transformed into instruction-following data as described in Section 3.1. During the feature alignment stage, we maintain the visual encoder and the LLM parameters in a frozen state, exclusively training the projection module. Additionally, this training phase involves fine-tuning token embeddings to accommodate the newly introduced visual codebook and two special tokens <vis> and </vis>. **(2) End-to-end instruction tuning**: In the second stage, we continue to keep the visual encoder frozen while concurrently training the projection module and LLM. This second stage leverages the entire open-ended instructions and contributes significantly to enhancing the model's capability of comprehending and effectively responding to complex multimodal instructions.

## 4 EXPERIMENTS

### 4.1 EXPERIMENTAL SETUP

**Implementation Details.** In this paper, we use the LLaMA-2 model (Touvron et al., 2023b) as the LLM backbone. Additionally, we use CLIP (Radford et al., 2021) as our visual encoder to achieve the best performance for non-visual generative tasks, and use VQ-GAN (Esser et al., 2021) as the default visual tokenizer for visual generation. The size of visual codebook $\mathcal{C}_v$ and language vocabulary is 8192 and 32000, respectively. In addition, we add <vis> and </vis> to the final unified codebook, indicating the starting and ending points of visual content. Similar to Liu et al. (2023), we construct 850K instruction-answer pairs for model training. Note that the model is trained to predict the agent's answer, and thus only sequence/tokens of answer will be used to compute the loss in the auto-regressive model. We also adopt LoRA (Hu et al., 2021) to reduce the computational cost for efficient tuning. We choose MineDojo (Fan et al., 2022) as the Minecraft platform to collect our instruction data and conduct experiments. Following Yuan et al. (2023), we use the environments of programmatic tasks to train basic policies with RL. These policies are trained to execute corresponding skills and keep fixed in all testing tasks.

**Evaluation Benchmarks.** We conduct experiments on three benchmarks to evaluate an agent's interaction ability in an open world. **(1) Environmental visual captioning (ENV-VC)**: given a snapshot, the model is asked to describe the agent's current status and environmental features from diverse aspects (e.g., life, food...). We evaluate the prediction's accuracy of each aspect by extracting corresponding answers from the output description to compare with the groundtruth. **(2) Foundational knowledge question answering (FK-QA)**: to assess the model's grasp of essential knowledge, we collect a set of 10,000 Minecraft-related questions from different sources, including the Wiki pages, Wiki tables, and Minecraft recipes. The performance is measured by the model's ability to provide correct answers to these questions. **(3) Skill prediction and planning (SPP)**: we utilize our proposed Steve-Eye to predict whether a skill has been successfully completed and assert its capability to generate executable high-level skill plans for long-horizon tasks.

**Table 1.** Comparisons of different model settings on the environmental visual caption benchmark. The experiments are conducted on 20K ENV-VC test set.

| Model | visual encoder | inventory 🗃 | equip 🪓 | object in sight 🐗 | life ❤️ | food 🍗 | sky 🟦 |
|-------|---------------|-----------|---------|------------------|--------|--------|--------|
| BLIP-2 | CLIP | 41.6 | 58.5 | 64.7 | 88.5 | 87.9 | 57.6 |
| Llama-2-7b | - | - | - | - | - | - | - |
| Steve-Eye-7b | VQ-GAN | 89.9 | 78.3 | 87.4 | 92.1 | 90.2 | 68.5 |
| Steve-Eye-13b | MineCLIP | 44.5 | 61.8 | 72.2 | 89.2 | 88.6 | 68.2 |
| Steve-Eye-13b | VQ-GAN | 91.1 | 79.6 | 89.8 | 92.7 | 90.8 | 72.7 |
| Steve-Eye-13b | CLIP | **92.5** | **82.8** | **92.1** | **93.1** | **91.5** | **73.8** |

**Table 2.** Comparisons of different data configurations on the environmental visual captioning benchmark, where "snapshot desc." denotes the 200K multimodal perception instruction dataset.

| | inventory 🗃 | equip 🪓 | object in sight 🐗 | life ❤️ | food 🍗 | sky 🟦 |
|---|-----------|---------|------------------|--------|--------|--------|
| no instruction tuning | 22.7 | 24.3 | 39.8 | 81.2 | 80.4 | 61.1 |
| w/o snapshot desc. | 46.2 (+23.5) | 40.9 (+16.6) | 41.2 (+1.4) | 83.0 (+1.8) | 82.4 (+2.0) | 63.3 (+2.1) |
| w/o icon images | 52.3 (+29.6) | 48.1 (+23.8) | 91.4 (+51.6) | 92.5 (+11.3) | 90.9 (+10.5) | 73.5 (+12.4) |
| full data | 92.5 (+69.8) | 82.8 (+58.5) | 92.1 (+52.3) | 93.1 (+11.9) | 91.5 (+11.1) | 73.8 (+12.7) |

## 4.2 ENVIRONMENTAL VISUAL CAPTIONING (ENV-VC)

We introduce this evaluation protocol for asserting Steve-Eye's multimodal perception function, which serves as an initial stride toward comprehensive evaluation of large multimodal models. Specifically, we collect 20,000 Minecraft snapshots (named ENV-VC test set) using Minedojo and apply the proposed data generation pipeline to create six questions for each snapshot, resulting in a total of 120K questions. These six questions pertain to the prediction of various aspects, including inventory items 🗃, equipment 🪓, objects in sight 🐗, life ❤️, food 🍗, and the visibility of sky 🟦.

During the inference phase, Steve-Eye predicts answers based on these questions and the input snapshot. Experimental results are presented in Table 1 and Table 2. As shown in Table 1, our visual encoder, when combined with multimodal instruction tuning, significantly enables the ability of the text-only language model LLM (Llama-2-7b) to comprehend the contents of the snapshots (Steve-Eye-7b). Notably, Steve-Eye outperforms BLIP-2 by a substantial margin due to the improved reasoning ability enabled by the larger LLM. Furthermore, the visual encoder plays a crucial role in facilitating multimodal understanding. Surprisingly, the model equipped with CLIP (Radford et al., 2021) surpasses the performance of the model using MineCLIP (Fan et al., 2022), achieving over +48.9%, +21.0% and +19.9% improvements in inventory, equipment, and object-in-sight predictions, respectively. We attribute this performance difference to the fact that MineCLIP does not prioritize fine-grained alignment during pre-training, despite being exposed to a diverse range of Minecraft videos. In summary, Steve-Eye's ability to comprehend visual cues from its surroundings lays the foundation for subsequent interactions with the world.

To investigate the effectiveness of various types of instructional data for multimodal perception, we carry out experimental comparisons with diverse data configurations in Table 2. First, our results showcase a significant improvement in the model's capacity to respond to instructional questions through instruction tuning, which leads to impressive gains of over +50% for inventory, equipment, and object-in-sight prediction. Furthermore, the inclusion of the multimodal perception dataset and icon images in the training data both contribute to a substantial improvement in the model's overall performance. Ultimately, the best results are achieved when combining all available data sources.

## 4.3 FOUDATIONAL KNOWLEDGE QUESTION ANSWERING (FK-QA)

Following Team (2022), we establish a question database specialized to assess our model's proficiency in generating responses pertaining to fundamental Minecraft knowledge. This evaluation is carried out through a validation dataset known as the FK-QA test set, which is further divided into two distinct subsets: TEXT and IMG. In the FK-QA TEXT subset, we generate a collection of 10,000 question-answer pairs curated from various sources, including the Minecraft-Wiki pages, Minecraft-Wiki tables, and Minecraft recipes. Each category comprises 2,000, 5,000, and 3,000 pairs, respectively. Upon receiving a response from Steve-Eye, we feed both the generated response

**Table 3.** Comparisons on FK-QA test set of the foundational knowledge question answering benchmark. The evaluation metrics consider both the scoring and accuracy dimensions simultaneously.

| | Scoring | | | | Accuracy | |
|---|---|---|---|---|---|---|
| | Wiki Page | Wiki Table | Recipe | TEXT All | TEXT | IMG |
| Llama-2-7b | 6.90 | 6.21 | 7.10 | 6.62 | 37.01% | - |
| Llama-2-13b | 6.31 (-0.59) | 6.16 (-0.05) | 6.31 (-0.79) | 6.24 (-0.38) | 37.96% | - |
| Llama-2-70b | 6.91 (+0.01) | 6.97 (+0.76) | 7.23 (+0.13) | 7.04 (+0.42) | 38.27% | - |
| gpt-turbo-3.5 | 7.26 (+0.36) | 7.15 (+0.94) | **7.97** (+0.87) | 7.42 (+0.80) | 41.78% | - |
| Steve-Eye-7b | 7.21 (+0.31) | 7.28 (+1.07) | 7.82 (+0.72) | 7.54 (+0.92) | 43.25% | 62.83% |
| Steve-Eye-13b | **7.38** (+0.48) | **7.44** (+1.23) | 7.93 (+0.83) | **7.68** (+1.06) | **44.36%** | **65.13%** |

and the corresponding groundtruth answer to ChatGPT. ChatGPT will first examine the accuracy of the response as a measure of answer correctness. To minimize variability in error, ChatGPT conducts a further evaluation, considering the response's accuracy, relevance, and level of detail. This comprehensive evaluation yields an overall score on a scale ranging from 0 to 10, where a higher score signifies superior overall performance. In the FK-QA IMG subset, we shift our focus to visual generation by employing 3,000 recipe images as groundtruth data. Here, our model is tasked with generating visual outputs for each item within the recipe inventory, following a specific order. The visual output is considered correct only if every element of the recipe is accurately generated. We adopt this metric to assert our model's ability to produce multimodal feedback.

Table 3 presents both scoring and accuracy results. It's worthy to note that Llama-2 exhibits consistent performance regardless of the model's scale, with Llama-2-70b only marginally outperforming the 7b-version by +1.26% in accuracy, meanwhile 13b-version performs even worse than 7b-version on the scoring results. We hypothesize that this phenomenon can be attributed to distinct variations in difficulty levels encountered within our FK-QA test set. Llama-2 fails to answer correctly for the challenging part regardless of its size due to essential knowledge missing. In contrast, Steve-Eye outperforms both Llama-2 and gpt-turbo-3.5, despite its considerably smaller scale. Furthermore, our model exhibits a more substantial improvement in responding to Recipe and Wiki Table questions as compared to Wiki Page questions. This disparity can likely be attributed to the fact that Wiki Page contains a large proportion of invalid questions (e.g., version, history), whereas Recipe and Wiki Table predominantly feature knowledge-related inquiries. Such result further validates the effectiveness of our approach in acquiring foundational knowledge. Unlike text-only LLMs, our model exhibits considerable ability to output visual contents, which achieves 65.13% accuracy on FK-QA IMG using the 13b-version. The multimodal generation ability enables Steve-Eye to better serve as an assistant for potential needed people such as beginners of this game. We show more details and cases in Appendix D.

## 4.4 Skill Prediction and Planning (SPP)

**Skill Prediction.** Similar to Section 3.1, we collect another 20K snapshot pairs in the form of $\{\mathcal{I}_0, \mathcal{I}_t\}$ from skill trajectories (referred to as Skill-Pred test). These pairs are input into our model to query the current execution status of the skill. The execution status can fall into one of three categories: success, failure, and running, with "running" signifying that the skill is currently in progress.

As shown in Table 4, our model exhibits commendable performance in skill status prediction. However, the performance is still far from enough to completely replace the rule-based game judgment adopted by the existing RL-based skill agents. These experiments indicate that, despite the excellent multimodal understanding capabilities of our model in open-world environments in previous experiments, it still falls short in fine-grained reasoning tasks that involve consecutive frames to some extent.

**Table 4.** Recall/Accuracy results on Skill-Pred test set for the skill prediction benchmark.

| | running (%) | success (%) | fail (%) |
|---|---|---|---|
| BLIP-2 | 65.2/58.8 | 49.8/54.3 | 42.1/51.8 |
| Steve-Eye-7b | 89.8/82.5 | 77.6/81.4 | 74.2/79.9 |
| Steve-Eye-13b | 92.1/84.2 | 80.5/83.1 | 76.8/81.5 |

**Skill Planning.** Following Yuan et al. (2023), we carry out evaluation on 24 difficult tasks in Minecraft. These tasks can be categorized into three types: cutting trees to craft primary items (7), mining cobblestones to craft advanced items (7), and interacting with mobs to harvest food and materials (10). Each task is tested for 30 episodes, where an episode refers to a multi-round interaction

**Table 5.** Comparisons on the skill planning benchmark. We test the mean success rates of all tasks, where each task is executed for 30 episodes using the same seeds for initialization.

| Model | | | | | | | | | | | | | | |
|---|---|---|---|---|---|---|---|---|---|---|---|---|---|---|
| MineAgent | 0.00 | 0.03 | 0.00 | 0.00 | 0.00 | 0.00 | 0.00 | 0.00 | 0.00 | 0.00 | 0.21 | 0.0 | 0.05 | 0.0 |
| gpt assistant | 0.30 | 0.17 | 0.07 | 0.00 | 0.03 | 0.00 | 0.20 | 0.00 | 0.20 | 0.03 | 0.13 | 0.00 | 0.10 | 0.00 |
| Steve-Eye-auto | 0.30 | 0.27 | 0.37 | 0.23 | 0.20 | 0.17 | 0.26 | 0.07 | 0.13 | 0.17 | 0.20 | 0.33 | 0.00 | 0.13 |
| Steve-Eye | **0.40** | **0.30** | **0.43** | **0.53** | **0.33** | **0.37** | **0.43** | **0.30** | **0.43** | **0.47** | **0.47** | **0.40** | **0.13** | **0.23** |

| Model | | | | | | | | | | |
|---|---|---|---|---|---|---|---|---|---|---|
| MineAgent | 0.46 | 0.50 | 0.33 | 0.35 | 0.0 | 0.0 | 0.06 | 0.0 | 0.0 | 0.0 |
| gpt assistant | 0.57 | 0.76 | 0.43 | 0.30 | 0.00 | 0.00 | 0.37 | 0.00 | 0.03 | 0.00 |
| Steve-Eye-auto | 0.70 | 0.63 | 0.40 | 0.30 | 0.17 | 0 | 0.37 | 0.03 | 0.07 | 0.00 |
| Steve-Eye | **0.73** | 0.67 | **0.47** | 0.33 | **0.23** | **0.07** | **0.43** | **0.10** | **0.17** | **0.07** |

process. At each round, the model receives the environmental feedback from the last round, plans a skill list based on the current status, and then picks up the top skill to execute. For each task episode, we set a maximum step between [3000, 10000]. In our evaluation, we compare Steve-Eye against two baseline approaches: (1) MineAgent (Fan et al., 2022), which completes tasks without decomposing them into basic skills, and uses PPO and self-imitation learning with CLIP reward, and (2) GPT Assistant, which employs ChatGPT as a high-level planner to generate skill plans by prompting itself with information from the environment and the agent's status. The results in Table 5 demonstrate that Steve-Eye significantly outperforms both baseline methods. Additionally, we conduct experiments in which Steve-Eye takes over the skill prediction function from the rule-based game judgment in Minecraft. This self-driven variant is referred to as 'Steve-Eye-auto.' Since the model's skill prediction is not always 100% accurate, Steve-Eye-auto does experience some performance degradation when compared to Steve-Eye. This degradation is more pronounced in longer, complex tasks (e.g., 🏹, ⛏, ⛏) as opposed to short-term tasks (e.g., 🟫, ●, 🛏). Nevertheless, Steve-Eye-auto still demonstrates significant performance improvements in most tasks, compared to the baselines. For additional details about this benchmark, please refer to Appendix C.3.

For better visualization, we provide a qualitative example of Steve-Eye completing the task "crafting stone axe with wooden pickaxe" as shown in Figure 5.

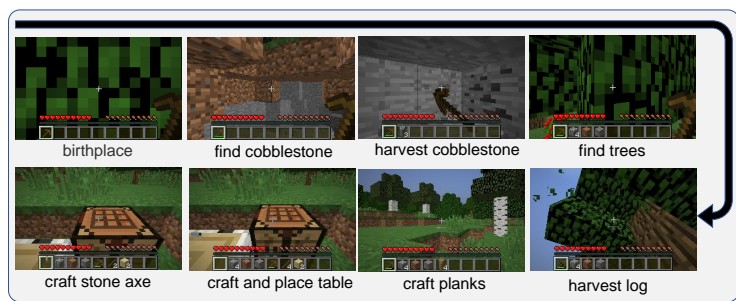

**Figure 5.** Snapshots of a qualitative example, illustrating how Steve-Eye completes the task of "crafting a stone axe with a wooden pickaxe." Our model generates a skill plan at each interaction round and selects the top skill from the plan list for execution.

## 5    CONCLUSION

In this paper, we explore enabling a large multimodal model to serve as a generative embodied agent in open worlds. We achieve this goal by proposing Steve-Eye, which combines the text-only language model with a visual encoder, allowing for a multimodal I/O interface to interact with the environment. With the help of ChatGPT, we curate questions to generate 850K instruction-following data to facilitate the agent's multimodal perception function, foundational knowledge mastery, as well as the capability of skill prediction and planning. Experiments on three open-world benchmarks verify the advantages of our Steve-Eye over a wide range of perspectives.

ACKNOWLEDGMENTS

This work was supported by NSFC under grant 62250068.

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

# Appendices

We offer a detailed description of the construction of our open-world instruction dataset, as outlined in Appendix A, including (1) multimodal perception instructions, (2) foundational knowledge instructions, (3) skill-related interaction instructions, and (4) template of instructional training data. Furthermore, we provide additional implementation details of our proposed benchmarks and conduct more analysis in Appendix B and Appendix C. In Appendix D, we present qualitative cases that illustrate our model's ability to provide intuitive visual feedback and serve as an intelligent chatbot with a multimodal input-output interface. Finally, in Appendix E, we offer further discussion of our work, such as the exploration of our model's potential applications in diverse environments.

## A    DATASET

In Minedojo, we set our agent's birthplace using a seed value of 500 during both data sampling and evaluation phases. This large seed value leads to spawning in a wide variety of locations within the map, including forests, plains, and lakes.

**Multimodal Perception Instruction.**    This dataset contains 200K instructional pairs. Figure 6 illustrates a partial listing of instructional questions employed for describing the content of the Minecraft snapshots. These instructions convey similar meanings, albeit with slight variations in natural language. Our generated responses to these instructions can be divided into two segments: (1) Responses directly derived from processed dense captions. To foster data diversity, we instruct ChatGPT to prompt a variety of description templates. Each caption is then rephrased by randomly selecting one of these templates, ensuring a broad spectrum of expressions. (2) Responses to the following questions about the caption. The model is required to accurately respond based on the selected question and the visual content of the caption.

> - "Describe the following Minecraft image in detail",
> - "Provide a detailed description of the given Minecraft image",
> - "Give an elaborate explanation of the Minecraft game image you see",
> - "Share a comprehensive rundown of the presented Minecraft image",
> - "Offer a thorough analysis of the Minecraft frame",
> - "Explain the various aspects of the Minecraft image before you",
> - "Examine the Minecraft image closely and share its details",
> - "Write an exhaustive depiction of the given Minecraft image"
> - "Clarify the contents of the displayed Minecraft image with great detail",
> - "Narrate the contents of the Minecraft image with precision"

**Figure 6.** Ten instruction examples for **multimodal perception instructions**.

**Foundational knowledge Instructions.**  The dataset comprises 250K training instances, which are organized into three distinct subsets: 200K icon image instructions, 10K recipe image instructions, and 40K Minecraft-Wiki corpus instructions. For the icon images, we generate questions aimed at prompting the model to recognize and describe item icons within the inventory, as depicted in Figure 7. Similarly, we curate instructional questions for recipe images as shown in Figure 8, with the objective of extracting information on completing specific recipes. In addition, we preprocess the raw Minecraft-Wiki HTML pages by removing irrelevant information (e.g., reference links) and unresolved data, transforming the raw corpus into a formatted, clean Markdown version. Leveraging the capabilities of ChatGPT, we employ this powerful language model to generate 10 questions, each with its corresponding answer, for every page of the cleaned Wiki corpus. This process yields a collection of 40K single-round question-answer pairs, which can be utilized for instruction tuning.

**Skill-related Interaction Instructions.**    For skill prediction, we utilize the skill policies trained by Yuan et al. (2023) to create a dataset comprising 200K skill trajectories. In each trajectory, we extract timestamps from the initial and $t$-th points to generate a snapshot pair, denoted as $\{\mathcal{I}_0, \mathcal{I}_t\}$.

- "Clarify the contents of the displayed inventory image with great attention to detail."
- "Characterize the inventory image with a meticulously detailed description."
- "Break down the individual slot elements within the inventory image with precision."
- "Take a step-by-step journey through the important details of the Minecraft inventory image."
- "Paint a vivid and descriptive narrative of the Minecraft inventory image."
- "Provide a precise narration of the contents within the Minecraft inventory image."
- "Thoroughly analyze the Minecraft inventory image in a comprehensive and detailed manner."
- "Illustrate the Minecraft inventory image through a descriptive and informative explanation."
- "Examine the Minecraft inventory image closely and share its intricate details."
- "Compose an exhaustive depiction of the given Minecraft inventory image."

**Figure 7.** Ten instruction examples of icon images for **foundational knowledge instructions**.

- "Provide a brief description of the given recipe image."
- "Offer a succinct explanation of the recipe picture presented."
- "Summarize the recipe content about icons of the image."
- "Give a short and clear explanation of the subsequent recipe image."
- "Share a concise interpretation of the recipe image provided."
- "Present a compact recipe description of the photo's key features."
- "Relay a brief, clear account of the recipe picture shown."
- "Render a clear and concise recipe summary of the photo."
- "Write a terse but informative recipe summary of the picture."
- "Create an icon narrative representing the recipe image presented."

**Figure 8.** Ten instruction examples of recipe image for **foundational knowledge instructions**.

- ""Steve is demonstrating his proficiency in {SKILL_NAME}, with the objective of achieving {SKILL_DEFINITION}. We'll now assess both the initial and current frames to determine if he has successfully executed the skill and the reasons behind it:"
- ""The skill Steve is performing is {SKILL_NAME}, and its intended outcome is to {SKILL_DEFINITION}. To determine whether Steve has accomplished this skill, we need to analyze both the starting and current frames:"
- ""Steve is currently engaged in executing {SKILL_NAME}, aiming to achieve {SKILL_DEFINITION}. In order to evaluate his success in performing this skill, we'll examine both the initial frame and the current frame:"
- ""The task at hand for Steve involves mastering {SKILL_NAME}, with the ultimate goal of accomplishing {SKILL_DEFINITION}. To ascertain whether he has successfully completed this skill, we'll analyze both the starting and current frames:"
- ""Steve is in the process of mastering the art of {SKILL_NAME}, with the specific objective of accomplishing {SKILL_DEFINITION}. We will now evaluate whether Steve has successfully executed this skill by comparing the initial and current frames:"
- ""The skill that Steve is currently executing is {SKILL_NAME}, and the intended outcome is {SKILL_DEFINITION}. To determine if Steve has effectively executed this skill, we'll assess both the initial frame and the current frame:"
- ""Steve is currently performing the {SKILL_NAME} skill, with the aim of achieving {SKILL_DEFINITION}. Let's analyze both the start frame and the current frame to determine whether he has succeeded and the reasons behind it:"
- ""Steve is demonstrating proficiency in the skill of {SKILL_NAME}, which is designed to accomplish {SKILL_DEFINITION}. Our evaluation will involve a comparison between the initial and current frames to assess the success of his execution:"
- ""The task Steve is undertaking is the mastery of {SKILL_NAME}, with the end goal of achieving {SKILL_DEFINITION}. To determine whether Steve has successfully executed this skill, we will analyze both the initial frame and the current frame:"
- ""Steve is in the process of executing the {SKILL_NAME} skill, with the ultimate aim of accomplishing {SKILL_DEFINITION}. Our assessment will involve a comparison between the starting frame and the current frame to determine if he has succeeded and why:"

**Figure 9.** Ten instruction examples for **skill prediction instructions**.

We then construct questions aimed at determining whether the agent successfully executed the skill or, in the case of failure, identifying the underlying reasons for the unsuccessful attempt. Illustrative examples of these skill prediction questions are provided in Figure 9. We also provide examples with such snapshot pairs in Figure 10.

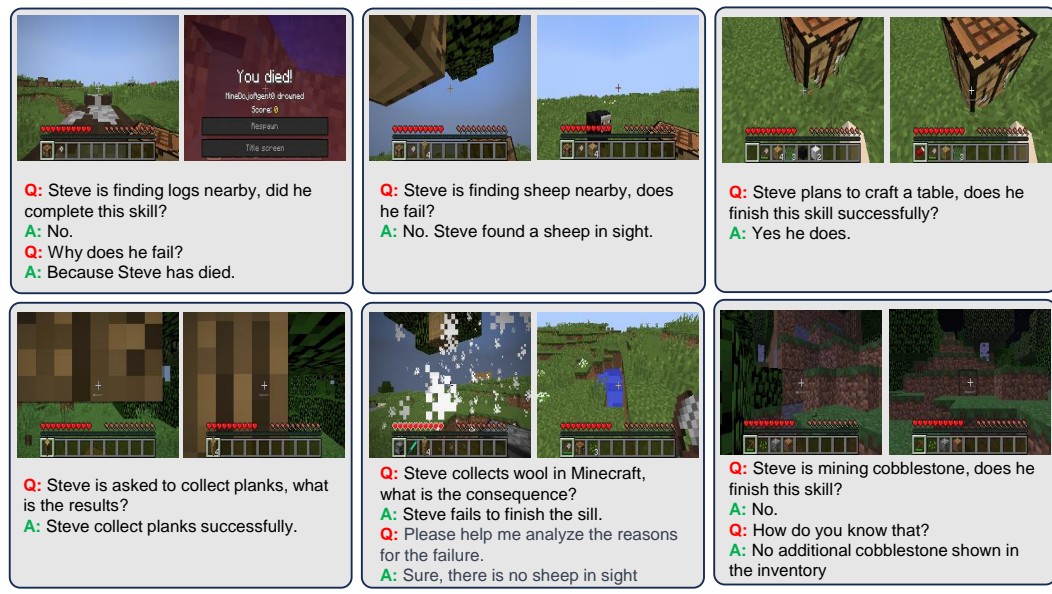

**Figure 10.** Illustrative examples of **skill prediction instruction** data with snapshot pairs.

**Template of Instructional Training Data.** Similar to Liu et al. (2023), we formulate each instructional instance as a multi-round conversation as shown in Figure 11, where $\mathcal{X}_{\text{head\_message}}$ is a sentence to describe this assistant (e.g., "You are in a chat between a curious human and an artificial intelligence assistant. You should serve as an assistant to give helpful, detailed, and polite answers to the human's questions."). The number of rounds relies on the input instruction content. And the input images (denoted as $<$ image $>$) will only be fed in the first round, while $\mathcal{X}_C$ may contain visual outputs with two additional tokens $<$ vis $>$ and $<$ /vis $>$.

> ### $\mathcal{X}_{\text{head\_message}}$ \n
>
> ### Human: $\mathcal{X}_Q^1$ <image> \n ### Embodied Agent: $\mathcal{X}_C^1$ \n
>
> ### Human: $\mathcal{X}_Q^2$ \n ### Embodied Agent: $\mathcal{X}_C^2$ \n
>
> ### Human: $\mathcal{X}_Q^3$ \n ### Embodied Agent: $\mathcal{X}_C^3$ \n
>
> ......

**Figure 11.** The unified template to generate input sequence for instructional tuning.

## B IMPLEMENTATION DETAILS OF BENCHMARK

**Foundational Knowledge Question Answering.** The 1000 evaluation QA pairs are generated by ChatGPT based on Minecraft knowledge from Wiki. It's important to note that these pairs are distinct from our instruction dataset and are not utilized in the pre-training phase to avoid data leakage. We will release them to ensure reproducibility. The evaluation process is cost-efficient, as the usage of gpt-turbo-3.5 charges only $0.0015$ per 1,000 tokens. Consequently, the entire evaluation costs less than 1 dollar.

**Skill Planning.** To clarify this benchmark, we begin by offering comprehensive task setup details in Table 6 and Table 7. During evaluation, we relocate the agent to a random location at the initiation of every episode, with distances of up to 500 units, ensuring that the agent spawns in an unfamiliar environment. Furthermore, for tasks that involve interacting with mobs, we enforce a maximum spawning distance of 30 units for cows and sheep.

**Table 6.** The setups of 24 tasks used in our skill planning evaluation, where "Initial Items" refers to the tools provided in the agent's inventory at the beginning of each episode, and "Max Steps" represents the maximum episode duration. Any episode exceeding this limit is classified as a task failure. The tasks are originally developed by Yuan et al. (2023).

**(a)** 7 tasks involving the process of "cutting trees to craft primary items".

| Task Icon | | | | | | | |
|---|---|---|---|---|---|---|---|
| Task Name | stick | crafting_table_nearby | bowl | chest | trap_door | sign | wooden_pickaxe |
| Initial Items | - | - | - | - | - | - | - |
| Max Steps | 3000 | 3000 | 3000 | 3000 | 3000 | 3000 | 3000 |

**(b)** 7 tasks involving the process of "mining cobblestones to craft advanced items".

| Task Icon | | | | | | | |
|---|---|---|---|---|---|---|---|
| Task Name | furname_nearby | stone_stairs | stone_slab | cobblestone_wall | lever | torch | stone_pickaxe |
| Initial Items | *10 | *10 | *10 | *10 | ↗ | *10 | ↗ |
| Max Steps | 5000 | 5000 | 3000 | 5000 | 5000 | 5000 | 10000 |

**(c)** 10 tasks involving the process of "interacting with mobs to harvest food and materials".

| Task Icon | | | | | | | | | | |
|---|---|---|---|---|---|---|---|---|---|---|
| Task Name | milk_bucket | wool | beef | mutton | bed | painting | carpet | item_frame | cooked_beef | cooked_button |
| Initial Items | , *3 | , *2 | ✗ | ✗ | , | , | | , ✗ | , ✗ | , ✗ |
| Max Steps | 3000 | 3000 | 3000 | 3000 | 10000 | 10000 | 3000 | 10000 | 10000 | 10000 |

**Table 7.** The setups of 10 long-horizon iron-based tasks, where "Initial Items" are provided in the agent's inventory at task beginning, and "Max Steps" refers to maximum environmental steps.

| Task icon | Task description | Initial tools | Max steps |
|---|---|---|---|
| | craft iron ingot | ↗*5, *64 | 8000 |
| | craft shears | ↗*5, *64 | 10000 |
| | craft bucket | ↗*5, *64 | 12000 |
| | craft iron pickaxe | ↗*5, *64 | 12000 |
| | craft iron axe | ↗*5, *64 | 12000 |
| | craft iron sword | ↗*5, *64 | 10000 |
| | craft iron shovel | ↗*5, *64 | 8000 |
| | craft tripwire hook | ↗*5, *64 | 8000 |
| | craft heavy weighted pressure plate | ↗*5, *64 t | 10000 |
| | craft iron trapdoor | ↗*5, *64 | 12000 |

For "*Steve-Eye*", it is rooted in a hierarchical framework to complete tasks. To be more specific, our model does not engage in predicting low-level moves. Instead, we implement low-level moves by additionally training an RL-based policy for each involved skill as introduced by Yuan et al. (2023). Then, our model exclusively generates high-level skill plans, delegating the actual skill execution to these pre-trained basic skill policies. Notably, we introduce a self-driven variant named "*Steve-Eye-auto*", which serves not only as a planner but also replaces the Minecraft rules to verify the successful execution of skills.

For the "*gpt-assistant*" baseline, we develop an interactive planning framework using ChatGPT. In this setup, ChatGPT serves as the planner, formulating skill plans in response to prompts that include task descriptions and environmental observations. This approach involves chain-of-thoughts

prompting, where we initially provide the planner with a few-shot demonstration with explanations to guide the initial planning step. To address frequent errors observed during testing, we incorporate specific planning rules into the prompts. As the planning process progresses, the planner encounters various scenarios: invalid skill names, redundant skills (already executed), successful skill execution, or failed skill execution. For each of these cases, we carefully craft language feedback and ask the planner to re-plan based on inventory changes.

For "*Plan4MC*", this approach utilizes a strategy similar to ours to break down a given Minecraft task into several skills and execute these skills one by one. By employing a specialized skill graph, Plan4MC guarantees the generation of executable plans with 100% certainty, thereby serving as an upper bound for our approach. As shown in the following, our Steve-Eye presents competitive performance when compared to Plan4MC, This is noteworthy considering Steve-Eye uses a significantly less optimal LLM (LLaMA-2) compared to ChatGPT used by Plan4MC.

## C  ADDITIONAL BENCHMARK ANALYSIS

### C.1  FOUNDATIONAL KNOWLEDGE QUESTION ANSWERING

It's important to acknowledge that the performance boost of this benchmark is limited by comparing Steve-Eye-13b and LlaMA-13b. We attribute this to the data noises within our Wiki data, particularly those extracted from Wiki Tables. As you can see in https://minecraft.fandom.com/wiki, Wiki data is highly structured, which inevitably causes data noises even with careful processing. To address this issue, we are dedicating more resources to refine the collected Wiki data for a better-quality version. Given that such an endeavor has not been previously undertaken, we believe that offering this knowledge-centric resource will significantly benefit the community.

### C.2  SKILL PREDICTION

In Minecraft, the rule-based analysis is beneficial since simply combining two adjacent frames may not provide enough information for skill assessment. Nevertheless, we do not directly employ it in our skill prediction evaluation. Instead, it plays an important role during the pre-training of the ENV-VC task, which is tailored to enable our model to interpret the current status primarily via visual cues. We believe relying solely on visual cues for predictions without explicit rule-based information is more in line with human's decision-making process, compared to a dependency on rule-based analysis, since there won't be explicit rules in our real life.

### C.3  SKILL PLANNING

We take a closer look at skill planning and identify several key factors contributing to the standout performance of Steve-Eye compared to other game agents like MineAgent:

**Adaptive Skill Planning Ability**: MineAgent employs a multi-task policy learned through RL, which can be seen as relying on static, predefined plans to accomplish each task. In contrast, Steve-Eye adopts a different approach by decomposing a task into individual executable skills, forming new skill plans based on current, real-time conditions. This method endows Steve-Eye with exceptional adaptability in various gaming environments, enabling it to devise more effective and executable skill plans. To validate this advantage, we conduct experiments (referred to as "ours w/o decomp") in Table 8, 9 and 10, which reveal a significant decline in performance without task decomposition. The results indicate that integrating task decomposition is beneficial for solving complex tasks with basic skills trained via RL.

**Additional Visual Understanding Ability.**   Our Steve-Eye directly leverages visual cues to assess its current status and respond accordingly. As shown in Table 5 in our main paper, visual cues empower Steve-Eye to outperform text-only models like "gpt-assistant" even based on a much lighter-weight LLM. Our work also suggests a universal application of visual cues in gaming agents, moving beyond the traditional reliance on rule-based indicators for status assessment.

**More Effective In-Context Reasoning Ability.**   As depicted in Table 3 in our main paper, Steve-Eye outperforms the vanilla Llama-2-7b by 6.4% on the foundational knowledge QA bench-

**Table 8.** Comparison of 7 tasks involving the process of "cutting trees to craft primary items."

| Model | stick | table | bowl | chest | trapdoor | sign | wooden_pickaxe |
|---|---|---|---|---|---|---|---|
| MineAgent | 0.00 | 0.00 | 0.00 | 0.21 | 0.0 | 0.05 | 0.0 |
| Plan4MC | 0.30 | 0.30 | 0.47 | 0.23 | 0.37 | 0.43 | 0.53 |
| w/o decomp | 0.03 | 0.03 | 0.23 | 0.13 | 0.03 | 0.07 | 0.00 |
| w/o ENV-VC | 0.27 | 0.27 | 0.33 | 0.43 | 0.37 | 0.27 | 0.40 |
| w/o FK-QA | 0.37 | 0.23 | 0.40 | 0.47 | 0.27 | 0.33 | 0.37 |
| ours | 0.40 | 0.3 | 0.43 | 0.53 | 0.33 | 0.37 | 0.43 |

**Table 9.** Comparison of 7 tasks involving the process of "mining cobblestones to craft advanced items."

| Model | furnace | stairs | stoneslab | stonewall | lever | torch | pickaxe |
|---|---|---|---|---|---|---|---|
| MineAgent | 0.00 | 0.03 | 0.00 | 0.00 | 0.00 | 0.00 | 0.00 |
| Plan4MC | 0.37 | 0.47 | 0.53 | 0.57 | 0.10 | 0.37 | 0.17 |
| w/o decomp | 0.13 | 0.23 | 0.17 | 0.13 | 0.03 | 0.07 | 0.00 |
| w/o ENV-VC | 0.23 | 0.30 | 0.33 | 0.37 | 0.27 | 0.23 | 0.23 |
| w/o FK-QA | 0.33 | 0.37 | 0.43 | 0.37 | 0.23 | 0.10 | 0.17 |
| ours | 0.30 | 0.43 | 0.47 | 0.47 | 0.40 | 0.13 | 0.23 |

**Table 10.** Comparison of 10 tasks involving the process of "interacting with mobs to harvest food and materials."

| Model | bucket | wool | beef | mutton | bed | paint | carpet | itemframe | beef | mutton |
|---|---|---|---|---|---|---|---|---|---|---|
| MineAgent | 0.46 | 0.50 | 0.33 | 0.35 | 0.0 | 0.0 | 0.06 | 0.0 | 0.0 | 0.0 |
| Plan4MC | 0.83 | 0.53 | 0.43 | 0.33 | 0.17 | 0.13 | 0.37 | 0.07 | 0.20 | 0.13 |
| w/o decomp | 0.50 | 0.43 | 0.23 | 0.13 | 0.03 | 0.03 | 0.17 | 0.03 | 0.00 | 0.03 |
| w/o ENV-VC | 0.67 | 0.43 | 0.33 | 0.27 | 0.13 | 0.07 | 0.27 | 0.07 | 0.23 | 0.03 |
| w/o FK-QA | 0.63 | 0.53 | 0.43 | 0.37 | 0.17 | 0.03 | 0.37 | 0.13 | 0.10 | 0.00 |
| ours | 0.73 | 0.67 | 0.47 | 0.33 | 0.23 | 0.07 | 0.43 | 0.10 | 0.17 | 0.07 |

**Table 11.** Comparison of 10 long-horizon iron-based tasks.

| Model | iron ingot | shears | iron bucket | iron pickaxe | iron axe | iron sword | iron shovel | tripwire hook | pressure plate | iron trapdoor |
|---|---|---|---|---|---|---|---|---|---|---|
| MineAgent | 0.00 | 0.00 | 0.00 | 0.00 | 0.00 | 0.00 | 0.00 | 0.00 | 0.00 | 0.00 |
| gpt-assistant | 0.20 | 0.00 | 0.00 | 0.03 | 0.00 | 0.00 | 0.03 | 0.00 | 0.03 | 0.00 |
| ours zero-shot | 0.17 | 0.10 | 0.13 | 0.07 | 0.03 | 0.03 | 0.27 | 0.03 | 0.00 | 0.00 |
| ours | 0.23 | 0.13 | 0.10 | 0.13 | 0.07 | 0.03 | 0.23 | 0.10 | 0.03 | 0.00 |

mark. Remarkably, it even exceeds GPT-Turbo-3.5, highlighting its acquisition of world knowledge. Therefore, Steve-Eye has enhanced adaptability to environmental changes when planning skills.

**Long-Horizon Planning Ability.** Long-horizon objectives in the skill planning benchmark (e.g., tasks of "mining cobblestones to craft stone pickaxe", "crafting bed" and "harvest cooked beef") typically can be broken down into dozens of skills and may require ten thousand steps to complete. Here, we conduct experiments on 10 long-horizon iron-based tasks in Table 11 and introduce their setup in Table 7. Steve-Eye consistently exhibits promising results across these long-horizon objectives, showing its robustness in handling complex game tasks. To further demonstrate Steve-Eye can learn game-playing strategies beyond the confines of the instruction-follow dataset, we demonstrate the results without the skill-related interaction instruction for training ("*ours zero-shot*"). Since the

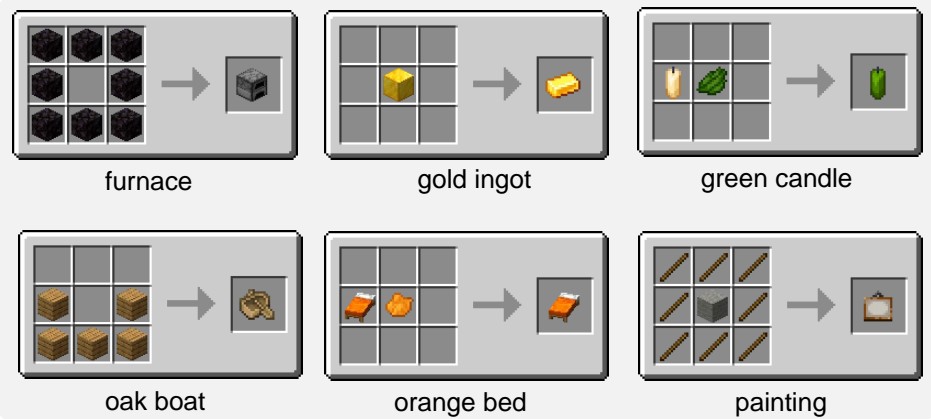

**Figure 12.** Qualitative examples of recipe image generation.

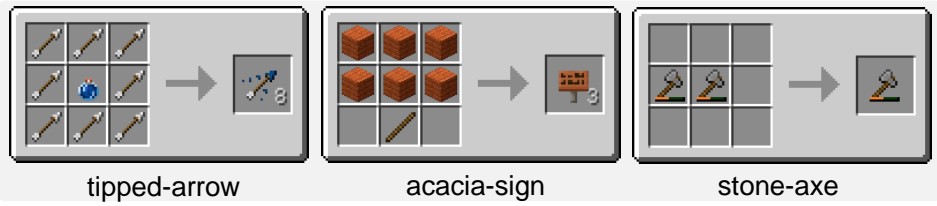

**Figure 13.** Illustrative examples of recipes that our model struggles to predict accurately. We attribute this failure to the complexities arising from fine-grained or semantically overlapping image information.

skill plans of these unseen tasks are not included in the training dataset, we believe these experiments effectively demonstrate Steve-Eye's ability beyond the scope of the collected instructions.

## D    QUALITATIVE RESULTS OF MULTIMODAL GENERATION

**Recipe Image Generation.**    Figure 12 showcases qualitative examples of our evaluation on the FK-QA IMG dataset. Utilizing a visual tokenizer like VG-GAN, our model demonstrates the ability to engage in visual generation, enabling it to provide visual feedback based on its comprehension of textual input. However, as shown in Figure 13, our model encounters difficulties when generating image content characterized by fine-grained or semantically overlapping elements. These challenges warrant further exploration in our future work.

**Multimodal ChatBot.**    In Figure 14, we present an overview of Steve-Eye functioning as a chatbot to receive task commands and execute them.

## E    FURTHER DISCUSSION

**Additional Open-World Exploration.**    In this paper, we have selected Minecraft as our open-world platform. Steve-Eye is not directly transferrable to other games due to each game's unique interface, especially for Minecraft known for its distinct pixel-style visuals. Nevertheless, Steve-Eye can be applied to other open-world environments, such as Virtual Home (Puig et al., 2018) and AI2THOR (Kolve et al., 2017), with minimal manual effort using the same methodology. These alternative benchmarks, when compared to Minecraft, exhibit a closer alignment with the real world. To some extent, this choice holds greater significance since our ultimate objective is to deploy the agent in the real world. To achieve this goal, we expand the Virtual Home benchmark by introducing a more extensive range of environments (50+ rooms), human-interaction tasks (200+ for each room), as well as diverse categories of actions (20+) and objects (100+), as illustrated in Figure 15. As a

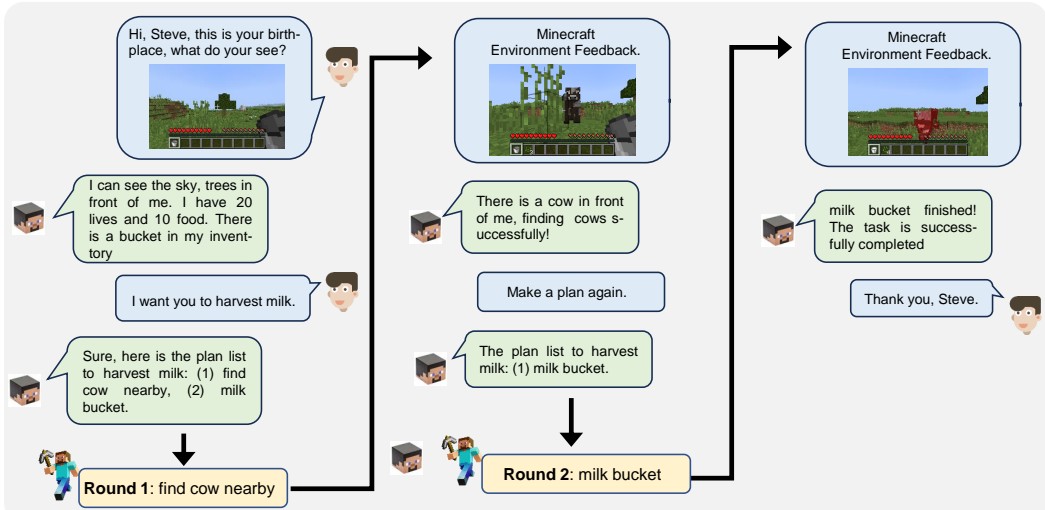

**Figure 14.** An overview of the process through which the chatbot receives and fulfills a task command.

simulated world, Virtual-Home demonstrates the possibility of seamlessly adapting Steve-Eye to a range of real-world simulation games. The corresponding validation and further exploration of open-ended embodied agents in a real-world context will be the focus of our future work.

**Association between ENV-VC and FK-QA for the Agent's Decision Making.** We verify this association can improve the agent's decision-making processes in Table 8, 9 and 10. When the ENV-VC task is removed, and we directly utilize visual cues without further instruction tuning, there's a notable drop in performance. This clearly demonstrates the critical role of the ENV-VC task in enhancing the model's ability to interpret and utilize visual cues effectively. Similarly, the exclusion of the FK-QA task yields a performance decrease, emphasizing the essential contribution of these two pre-training tasks in enhancing our model's decision-making capabilities.

**Comparison with more advanced methods like Voyager.** We choose not to compare Steve-Eye with methods like Voyager, as there are significant differences in their underlying mechanisms. Voyager controls the agent by generating code for the Mineflayer API using GPT-4, a process that is inherently different from Steve-Eye, which is driven by a considerably lighter-weight LLM and employs pre-defined RL policies for each skill control. Additionally, Voyager heavily relies on curated prompting for its planning and skill selection process, a feature not as prominently utilized in our approach. Given these fundamental differences, a direct comparison between Steve-Eye and Voyager would not be equitable. However, it's crucial to recognize that Steve-Eye is compatible with Voyager and other similar methods. Voyager uses LLMs to translate an agent's observations into text, while Steve-Eye excels at interpreting multimodal cues. This capability positions Steve-Eye as a potential multimodal foundation model to replace the LLMs in methods like Voyager.

**Implementation of a Fully Embodied Agent.** A fully embodied agent trains the model to predict the next move from videos and instructions, which could perform better compared with our existing hierarchical framework. However, our initial attempt to predict moves from videos and instructions has proven to be a substantial challenge and did not yield satisfactory results before. As a result, currently, our Steve-Eye assesses only the success or failure of skill execution at each step, so that Steve-Eye can decide whether to re-plan. Nevertheless, we agree that directly predicting low-level moves based on visual cues could be more advantageous for creating realistic embodied agents. Thus, we are committed to exploring this direction in our future work.

**Clarification on Image Output Utilization.** Firstly, integrating visual generation into our pre-training introduces a novel task that significantly broadens the scale and diversity of our instruction-following dataset. This expansion helps prevent oversimplification of the training process and ensures that the model consistently produces accurate and coherent responses, especially when dealing with complex environments. Secondly, as discussed in our introduction, we argue that multimodal

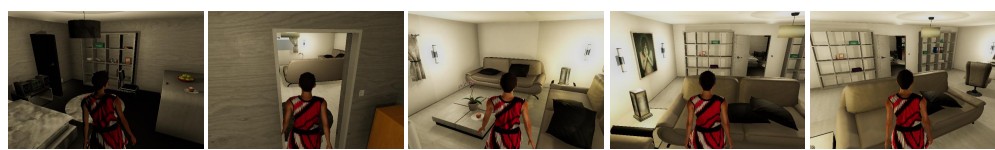

**Task Name**: relax on sofa

**Task Description**: I go to the living room and sit in the sofa

**Task Plan**: {"1": walk living room, "2": walk couch, "3": find couch, "4": walk couch, "5": sit couch}

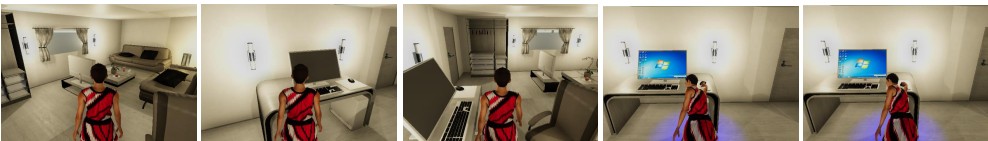

**Task Name**: browse the Internet

**Task Description**: I go to the office and sit in a chair, I turn on the computer and grab the mouse. I type on the keyboard and starting working on the computer.

**Task Plan**: {"1": walk living-room, "2": walk desk, "3": find desk, "4": find chair, "5": sit chair, "6": find computer, "7": switch-on computer, "8": find mouse, "9": grab mouse, find keyboard, "10": type keyboard, "11": turn-to computer, "12": look-at computer}

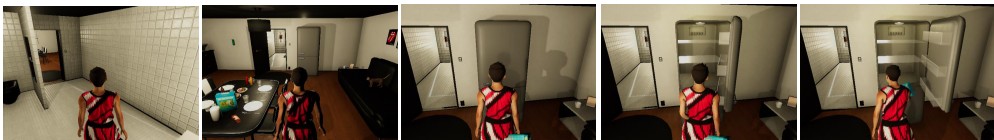

**Task Name**: put milk in the freezer

**Task Description**: I walk into kitchen, look for the milk, walk to milk, look for refrigerator, walk to refrigerator, open door, put the milk in the refrigerator

**Task Plan**: {"1": walk dining-room, "2": walk milk, "3": find milk, "4": turn-to milk, "5": grab milk, "6": look-at freezer, "7": walk freezer, "8": open freezer, "9": put milk

**Figure 15.** Task examples from the extended Virtual-Home benchmark, where elements in green, cyan and red represent action, room, and object categories, respectively. Our benchmark includes a diverse range of tasks that simulate interactions between individuals and their room environments. It contains over 50 distinct room setups, involving 20 unique actions, and 100 objects. Each room presents a selection of more than 200 distinct tasks.

outputs offer a more intuitive way to represent the world, leading to an effective interface for interaction with human users. Therefore, although the multimodal output in this work is not generated for specific tasks, we believe that this is crucial for developing a general-purpose game assistant.

