# OpenReview forum: "Steve-Eye: Equipping LLM-based Embodied Agents with Visual Perception in Open Worlds"
_ICLR.cc/2024/Conference — ICLR 2024 poster_

### Official Review · Reviewer_mQD9 · 2023-10-27

**Soundness:** 3 good
**Presentation:** 3 good
**Contribution:** 2 fair
**Rating:** 6
**Confidence:** 3

**Summary:**

This paper proposes a new end-to-end architecture for multimodal agent interacting with a simulated environment. Specifically, it proposes to fuse a visual encoder with a pre-trained large language model for visual and textual generation. The model is trained on a curated dataset of Minecraft data, and is able to achieve convincing results. Overall, the work attempts to enhance interaction the capability of text-only LLM interaction for embodied agents and the results are promising for future work.

**Strengths:**

The proposed architecture is intuitively sound. It seems to be the natural way to merge a visual encoder, an LLM, and an image generator in an end-to-end pipeline.

The paper contains some smart ways to curate a large dataset.

The author suggests to publish codes and projects for reproducibility, which is nice.

**Weaknesses:**

The tasks are still fairly constraint in that everything is image-text or text-text pairs for evaluation. The task does not really test on the agent’s ability to navigate or perform tasks in Minecraft environment directly.

(FK-QA) Evaluation is still focusing on in-domain knowledge about Minecraft-related questions. There seems to have no procedures or details specifying the methods to ensure data leakage. Especially, given that the LLM is selected as a LLAMA2, and directly evaluate on data procured from the internet. There could be an additional benchmark on using LLAMA to fine-tune on the text-only datasets and to evaluate on the same sets of text-only evaluation set.

Table-3 has Llama 2-7b model outperforming Llama 2-13b model. What would be a good explanation for this behaviour?

**Questions:**

See weaknesses.

---

> ### Author Response · Authors · 2023-11-21
> **Responses to Reviewer mQD9**
>
> Thank you for your valuable suggestion.
>
> # 1. Does task include test on the agent's ability?
> Yes, our skill planning benchmark involve the tasks performing in Minecraft environment directly, following another Minecraft game agent work: Plan4MC.
> In this benchmark, our model is required to make an initial plan for a given task, adjust the plan as the agent encounters varied scenario: invalid skill names, redundant skills (already executed), successful skill execution, or failed skill execution.
> After skill planning, Steve-Eye execute each skill using its pre-trained low-level policies.
> The experiments related to skill planning are detailed in Section 4.4 of our main paper.
> Appendix 2 provides an in-depth look at each task.
> We have also conducted further experiments on long-horizon tasks, the results of which are presented in **Table R4** in the response to Reviewer k9Af.
> These experiments  demonstrates the effectiveness of our Steve-Eye in performing tasks within the Minecraft environment.
>
> # 2. Questions about FK-QA Evaluation
> Firstly, the QA pairs used for FK-QA evaluation are curated by ChatGPT.
> In order to avoid data leakage, we ensure the question prompt varies from the instruction pairs for pre-training and will not be used during pre-training.
>
> Secondly, we have carried out experiments using a text-only benchmark with the finetuned LLaMA model.
> As anticipated, the finetuned LLaMA demonstrates lower performance compared to its original version, aligning with findings from prior research.
> However, given our objective to develop an agent proficient in games like Minecraft, we consider this performance decline acceptable. This trade-off is deemed worthwhile to achieve enhanced capabilities in our target domain for general-purpose game agents.
>
>
> # 3. Why does LLaMA-2-7b outperform LLaMA-2-13b in Table.3?
> In our observations, the LLama-2-13b model does not exhibit consistent improvements over its 7b predecessor.
> This trend is not isolated; similar outcomes are evident in other instruction-tuning practices based on LLaMA, including LLaVA [2].
> By contrast, the LLaMA-2-70b model generally demonstrates notable enhancements when compared to both its 7b and 13b counterparts, across nearly all benchmarks.
>
> [2] Visual Instruction Tuning

---

> > ### Author Response · Authors · 2023-11-22
> > **We sincerely hope that our rebuttal has addressed your concerns. Could you please confirm this?**
> >
> > Dear Reviewer mQD9, thanks for your time in reviewing our work. As the author-reviewer discussion period is expected to end on tomorrow (Nov 22nd), we wonder if you can kindly check whether our rebuttal resolves your concerns.
> >
> > We are more than happy to have further discussions with the reviewer if there are any remaining issues.

---

### Official Review · Reviewer_k9Af · 2023-10-31

**Soundness:** 3 good
**Presentation:** 3 good
**Contribution:** 3 good
**Rating:** 5
**Confidence:** 3

**Summary:**

This paper proposed Steve-eye, a multi-modal LLM for visual-based embodied game playing in the simulated environment. Especially,
as previous game agent mainly obtain environment informations directly from game engine, Steve-eye want to intergrate perception system into game playing agent. To achieve this objective, this paper construct a multi-modal instruction dataset for instruction-tuning. And, the paper construct some benchmark to quantitatively evaluate the different capability of the Steve-eye.

**Strengths:**

1. The paper dedicate to a interesting and valuable problem that make game agent more human-like (use visual to sense the world).
2. Steve-eye divided the visual perception for several ends: generation, QA, caption, skill planning, which suit the minecraft game well.
3. Generate a large multi-modal instruction dataset for this task.

**Weaknesses:**

1. The most important question for Steve-eye is the evaluation. Steve-eye is not evaluated on some long-horizon objectives like Voyager and GITM. With generated multi-modal instruction dataset, it is so intuitive that the model can perform well on the ENV-VC, FK-QA, SPP benchmarks as the instructions are generated for these tasks. The skill plan, visual caption or QA are actually studied on real-world images by some VLLM such as LLaVA. There is no reason to go back and evaluate VLLM on simulation environment just for such tasks.
I think the main reason for the community to carry on research on game engine is to evaluate much more complex tasks that are not hard to attempt in real world. In this regard, I think the Steve-eye should have some evaluation manners as Voyager and GITM, to show what happens when we replace the environment information of game engine to visual perception.

2. Is the model trained on such instruction dataset generalizable to new game? Such as GTAV?

**Questions:**

No anymore questions.

---

> ### Author Response · Authors · 2023-11-21
> **Responses to Reviewer k9Af**
>
> Thank you for your valuable suggestion, particularly regarding the supplementation of long-horizon tasks.
>
> # 1. Long-horizon Evaluation Objectives
>
> In fact, our work does involve long-horizon objectives in the skill planning benchmark (e.g., tasks of ``mining cobblestones to craft stone pickaxe'', ``crafting bed'' and ``harvest cooked beef'').
> These tasks are typically broken down into dozens of skills and may require ten thousands of steps to complete.
> More details of these long-horizon tasks are provided in **Table 6 in Appendix.2**.
> In addition to these tasks, we conduct experiments on 10 new long-horizon objectives.
> These objectives are iron-based task as detailed in **Table 7 in the Appendix.2**.
> As shown in **Table.R4** , our Steve-Eye consistently exhibits promising results across these long-horizon objectives, showing its robustness in handling complex game tasks.
> In addition, we provide addition experiments in **Table R1, R2, R3** (in reponses to ijGQ) to demonstrate that visual perception is beneficial to text-only game engines.
>
> **Table R4**: 10 long-horizon iron-based tasks.
> | Model        | iron ingot | shears | iron bucket | iron-pickaxe | iron-axe | iron-sword | iron-shovel | tripwire hook | pressure plate | iron trapdoor |
> |--------------|------------|--------|-------------|--------------|----------|------------|-------------|---------------|----------------|---------------|
> | MineAgent    | 0.00       | 0.00   | 0.00        | 0.00         | 0.00     | 0.00       | 0.00        | 0.00          | 0.00           | 0.00          |
> | gpt-assistant| 0.20       | 0.00   | 0.00        | 0.03         | 0.00     | 0.00       | 0.03        | 0.00          | 0.03           | 0.00          |
> | ours         | 0.23       | 0.13   | 0.10        | 0.13         | 0.07     | 0.03       | 0.23        | 0.10          | 0.03           | 0.00          |
>
>
> # 2. Is the model trained on such instruction dataset generalizable to new game? Such as GTAV?
> Our Steve-Eye is not directly transferrable to other games due to each game's unique interface, especially for Minecraft known for its distinct pixel-style visuals.
> Nonetheless, the overall framework of Steve-Eye, including the pre-training tasks, the data collection process, and model structure, is highly adaptable and can be tailored for new games.
> This significantly simplifies the process of applying Steve-Eye to various open-world games.
> In fact, we are exploring to extend this framework to other open-ended environments, such as Virtual-Home as detailed in Appendix 4.
> As a simulated world, Virtual-Home demonstrates the possibility of seamlessly adapting our Steve-Eye to a range of real-world simulation games, including GTAV you mentioned.

---

> > ### Author Response · Authors · 2023-11-22
> > **We sincerely hope that our rebuttal has addressed your concerns. Could you please confirm this?**
> >
> > Dear Reviewer k9Af, thanks for your time in reviewing our work. As the author-reviewer discussion period is expected to end on tomorrow (Nov 22nd), we wonder if you can kindly check whether our rebuttal resolves your concerns (e.g, long-horizon evaluation objectives).
> >
> > We are more than happy to have further discussions with the reviewer if there are any remaining issues.

---

> ### Comment · Reviewer_k9Af · 2023-11-22
> **Response to author**
>
> The evaluation provided by the author for skill planning doesn't yet meet the long-horizon criteria for me. As mentioned in my comment, I am looking forward to see evaluation similar to voyager and GITM [A], such as the number of items unlocked, the extent of exploration range.
>
> Again, as the author constructs instruction-follow dataset to train the model on those skills, it is intuitive that steve-eye can do well on them. All the evaluation is still under the generated instruction-follow dataset.
> I believe further evaluation is necessary to demonstrate the agent's capability to learn game-playing strategies beyond the confines of the instruction-follow dataset. Moreover, it would be valuable to see if the agent can generalize its skills to other games, given that creating instruction-follow datasets for each game is not a feasible approach.
>
> Although my concerns remain unaddressed, I acknowledge the pioneering nature of this work in the field and will maintain my rating accordingly.
>
> [A]. Ghost in the Minecraft: Generally Capable Agents for Open-World Environments via Large Language Models with Text-based Knowledge and Memory

---

> > ### Author Response · Authors · 2023-11-22
> > **Additional Responses to Reviewer k9Af**
> >
> > Thank you for your further reply!
> > In response to your concerns, we have conducted additional analysis and hope that the following detailed response will address your questions:
> >
> > # 1. Evaluation on metrics like Voyager and GITM
> > We understand your concern about our evaluation for skill planning. However, we respectfully argue that the evaluation of Steve-Eye for **your long-horizon criteria** is beyond the focus of our paper.
> >
> > As we claim our contributions in the introduction section are threefold: instruction dataset, large multimodal model and training, and benchmarks including environmental visual captioning, foundational knowledge question answering, and skill prediction and planning. We can see that skill planning is a small fraction of our contributions. For Voyager and GITM, their work mainly focuses on skill planning, and they leverage the much more powerful GPT-4. Thus, they evaluate "the number of items unlocked, the extent of exploration range." However, this is not the case for Steve-Eye, since it is merely based on Llama-2-13B. Moreover, for skills, Voyager uses the generated code for Mineflayer API as skills, GITM uses manually structured skills, and Steye-Eye uses RL-learned policies as skills. Thus, it is unfair to evaluate Steye-Eye on their metrics, and we honestly admit that our current Steve-Eye will certainly underperform Voyager and GITM there. *In short, we believe our evaluation supports our claim on skill planning and it is unnecessary to evaluate Steve-Eye on your long-horizon criteria*.
> >
> > # 2. Concern about evaluation which is still under the generated instruction-follow dataset.
> >
> > Firstly, it's important to mention that we set our agent's birthplace in Minedojo using a seed value=500 during both data sampling and evaluation phases.
> > The large seed value leads to spawning in a wide variety of locations within the map, including forests, plains, and lakes.
> > Due to the diverse range of birthplace, our instruction-following dataset is relatively limited in scale, meaning it is far enough from including all possible, unseen scenarios.
> >
> > Secondly, for the FK-QA benchmark, the evaluation QA pairs are generated by ChatGPT based on Minecraft knowledge from Wiki.
> > It's important to note that these pairs are distinct from our instruction dataset and are not utilized in the pre-training phase.
> >
> > Lastly, to further demonstrate Steve-Eye can learn game-playing strategies beyond the confines of the instruction-follow dataset.
> > As shown in Table R5, we carry out additional experiments on 10 long-horizon iron-based tasks without their skill-related interaction instruction for training (ours zero-shot).
> > Since the skill and plans of these unseen tasks are not included in the training dataset, we believe these experiments effectively demonstrate Steve-Eye's ability beyond the scope of the collected instructions.
> > Although performing worse, we observe that our Steve-Eye can still complete these challenging tasks and outperforms text-only methods driven by ChatGPT (gpt-assistant).
> >
> > Table R5: 10 long-horizon iron-based tasks.
> > | Model           | iron ingot | shears | iron bucket | iron-pickaxe | iron-axe | iron-sword | iron-shovel | tripwire hook | pressure plate | iron trapdoor |
> > |--|---|---|----|----|-----|-----|---|--|-|----|
> > | MineAgent    | 0.00       | 0.00   | 0.00   | 0.00  | 0.00     | 0.00       | 0.00        | 0.00   | 0.00    | 0.00          |
> > | gpt-assistant   | 0.20       | 0.00   | 0.00    | 0.03    | 0.00     | 0.00       | 0.03   | 0.00  | 0.03   | 0.00          |
> > | ours zero-shot  | 0.17       | 0.10   | 0.13   | 0.07 | 0.03     | 0.03       | 0.17    | 0.03   | 0.00      | 0.00          |
> > | ours            | 0.23       | 0.13   | 0.10        | 0.13  | 0.07     | 0.03       | 0.23    | 0.10       | 0.03   | 0.00          |
> >
> >
> > # 3. Can the agent generalize its skills to other games?
> > We believe the framework of our Steve-Eye  can be transferred to other games with much less effort, especially when compared with methods like Voyager or GITM.
> > A key strength of our approach lies in its use of image-text pairs to construct instructional data, which can be easily obtained from the Internet.
> > Given the vast amount of game tutorial videos on platforms like YouTube, adapting Steve-Eye for training in different gaming environments appears highly feasible.
> >
> > Unlike Steve-Eye, methods like Voyager or GITM face the challenge of needing to completely rewrite their prompting systems to align with the unique mechanics of each new game.
> > This requirement significantly hinders their adaptability.
> >
> > Therefore, the relative ease of adapting Steve-Eye to diverse games stands out as a key benefit of our approach, making it more versatile and user-friendly in a variety of game scenarios.
> > We are progressing towards realizing this adaptability by implementing our approach in realistically simulated environments, such as Virtual-Home.
> >
> > Happy to have further discussions with the reviewer if there are any remaining issues.

---

### Official Review · Reviewer_ijGQ · 2023-11-01

**Soundness:** 3 good
**Presentation:** 3 good
**Contribution:** 3 good
**Rating:** 5
**Confidence:** 4

**Summary:**

The "Steve-Eye" paper introduces embodied game-playing agents with visual perception capabilities and the potential to interact intelligently in open-world environments. By integrating large language models with a visual encoder, the paper enables agents to represent their surroundings and comprehend critical knowledge in the environment in a more intuitive manner. Utilizing Minecraft as a primary evaluation platform, the model demonstrates its effectiveness in a complex and dynamic setting, showcasing its versatility and potential as a general-purpose assistant.

**Strengths:**

**1. Strong Motivation:** The paper identifies and addresses a critical challenge in the field of embodied agents, highlighting the limitations of text-only interactions and demonstrating the necessity of incorporating visual perception for a more intuitive and effective user experience.

**2. Extensive Task Definitions:** The authors have meticulously defined a variety of tasks to evaluate the agent’s performance, spanning across environmental visual captioning, foundational knowledge question answering, and skill prediction and planning. This comprehensive approach ensures a thorough assessment of the agent’s capabilities, showcasing its versatility and adeptness in handling different aspects of open-world interaction.

**3. Large and Rich Dataset:** The paper introduces an extensive open-world instruction dataset, specifically curated to train the Steve-Eye model. This dataset can be a good contribution to the field for future instruction tuning of large multimodal models.

**Weaknesses:**

**1. In-Depth Analysis on Skill Planning Required:** The paper presents Steve-Eye, an innovative integration of visual perception with Large Language Models (LLMs) for embodied agents. However, there’s a discernible lack of comprehensive discussion and analytical depth in the aspect of skill planning, particularly when compared to other game-playing agents such as “MineAgent.” While Steve-Eye showcases commendable performance, a more exhaustive exploration of its strategic capabilities, especially in comparison to MineAgent, is crucial. This detailed analysis could unveil the underlying reasons behind Steve-Eye’s superior performance, providing readers with substantial insights and a clearer understanding of its capabilities in the embodied task of planning.

On a related note, the paper discusses two other tasks: Environmental Visual Captioning (ENV-VC) and Foundational Knowledge Question Answering (FK-QA). While these tasks are undoubtedly interesting and add value to the paper, their direct connection to the agent’s in-game execution and decision-making processes is not explicitly clear. Strengthening this connection and elaborating on how these tasks intricately weave into the agent’s planning and action sequences would significantly enhance the paper’s overall contribution to the field.

**2. Baseline Selection Could Be Improved:** The inclusion of “gpt-assistant” as a baseline in the performance evaluation, particularly noted in Table 5, brings about certain ambiguities. The choice of baselines is critical, and in this context, one might wonder if “Voyager” or other models specifically tailored for the Minecraft environment would serve as more apt comparisons. By opting for baselines that are more closely aligned with the Minecraft gaming milieu, the paper could present a more robust and convincing argument for Steve-Eye’s effectiveness.

**3. Clarification on Image Output Utilization Needed:** As depicted in Figure 4 and discussed in various sections of the paper, Steve-Eye has the capability to generate image outputs. However, the practical application and utility of these image outputs in the context of the tasks at hand appear somewhat nebulous. Providing readers with a clearer exposition on how these image outputs can be leveraged for specific tasks would enhance the paper’s clarity and comprehensibility.

Minor:
**4. Figure 1 Requires Better Visual Clarity:** My initial interaction with Figure 1 was marked by confusion, necessitating several revisits to fully grasp its intended message. A visual element such as a horizontal separator line could significantly improve the figure’s readability, distinctly delineating the unfavorable responses from the preferable ones. Currently, the figure could be misinterpreted as a continuous chat, leading to potential misunderstanding. Implementing this small yet impactful change would streamline the reader’s experience, fostering a quicker and clearer comprehension of the depicted scenarios.

Minor:
5. Figure 7/8/12 in the appendix are not in the correct order.

**Questions:**

See above

---

> ### Author Response · Authors · 2023-11-21
> **Responses to Reviewer ijGQ**
>
> # 1. In-Depth Analysis on Skill Planning Required
>
> Thank you for your valuable suggestions!
> In this refined analysis, we take a closer look at skill planning and identify several key factors contributing to the standout performance of Steve-Eye compared to other game agents like MineAgent:
>
> * **Adaptive Skill Planning Ability**:
> MineAgent employs a multi-task policy learned through RL, which can be seen as relying on static, predefined plans to accomplish each task.
> In contrast, Steve-Eye adopts a different approach by decomposing a task into individual executable skills, forming new skill plans based on current, real-time conditions.
> This method endows Steve-Eye with exceptional adaptability in various gaming environments, enabling it to devise more effective and executable skill plans.
> To validate this advantage, we conduct experiments (referred to as "ours w/o decomp") in **Tables R1, R2, and R3** below, which reveal a significant decline in performance without task decomposition.
> The results indicates that integrating task decomposition is beneficial for solving complex tasks with basic skills trained via RL.
>
> * **Additional Visual Understanding Ability**:
> Our Steve-Eye directly leverages visual cues to assess its current status and respond accordingly.
> As shown in **Table 5** in our main paper, visual cues empower Steve-Eye to outperforms text-only models like ``gpt-assistant'' even based on a much ligher-weight LLM.
> Our work also suggests a universal application of visual cues in gaming agents, moving beyond the traditional reliance on rule-based indicators for status assessment.
>
> * **More Effective In-context Reasoning Ability**:
> As depicted in **Table 3** in our main paper, Steve-Eye outperforms the vanilla Llama-2-7b by 6.4\% on the foundational knowledge QA benchmark.
> Remarkably, it even exceeds GPT-Turbo-3.5, highlighting its acquisition of world knowledge.
> Therefore, Steve-Eye has enhanced adaptability to environmental changes when planning skills.
>
> **Table R1**: 7 tasks involving the process of ``cutting trees to craft primary items''
> | Model      | stick | table | bowl | chest | trapdoor | sign | wooden pickaxe |
> |------------|-------|-------|------|-------|----------|------|----------------|
> | MineAgent  | 0.00  | 0.00  | 0.00 | 0.21  | 0.0      | 0.05 | 0.0            |
> | Plan4MC    | 0.30  | 0.30  | 0.47 | 0.23  | 0.37     | 0.43 | 0.53           |
> | ours w/o decomp | 0.03  | 0.03  | 0.23 | 0.13  | 0.03     | 0.07 | 0.00           |
> | ours w/o ENV-VC | 0.27  | 0.27  | 0.33 | 0.43  | 0.37     | 0.27 | 0.40           |
> | ours w/o FK-QA  | 0.37  | 0.23  | 0.40 | 0.47  | 0.27     | 0.33 | 0.37           |
> | ours       | 0.40  | 0.30  | 0.43 | 0.53  | 0.33     | 0.37 | 0.43           |
>
>
> **Table R2**: 7 tasks involving the process of ``mining cobblestones to craft advanced items''
> | Model     | furnace | stonestairs | stoneslab | cobblestonewall | lever | torch | stonepickaxe |
> |--|-|-|-|---|-------|-------|---------|
> | MineAgent | 0.00    | 0.03   | 0.00      | 0.00      | 0.00  | 0.00  | 0.00    |
> | Plan4MC   | 0.37    | 0.47   | 0.53      | 0.57      | 0.10  | 0.37  | 0.17    |
> | ours w/o decomp| 0.13    | 0.23   | 0.17      | 0.13      | 0.03  | 0.07  | 0.00    |
> | ours w/o ENV-VC| 0.23    | 0.30   | 0.33      | 0.37      | 0.27  | 0.23  | 0.23    |
> | ours w/o FK-QA | 0.33    | 0.37   | 0.43      | 0.37      | 0.23  | 0.10  | 0.17    |
> | ours      | 0.30    | 0.43   | 0.47      | 0.47      | 0.40  | 0.13  | 0.23    |
>
> **Table R3**: 10 tasks involving the process of ``interacting with mobs to harvest food and materials''
> | Model     | milkbucket | wool | beef | mutton | bed | painting | carpet | itemframe | cookedbeef | cookedmutton |
> |-|-|-|-|-|-|-|-|-|-|-|
> | MineAgent | 0.46   | 0.50 | 0.33 | 0.35   | 0.0 | 0.0   | 0.06   | 0.0       | 0.0  | 0.0    |
> | Plan4MC   | 0.83   | 0.53 | 0.43 | 0.33   | 0.17| 0.13  | 0.37   | 0.07      | 0.20 | 0.13   |
> | ours w/o decomp| 0.50   | 0.43 | 0.23 | 0.13   | 0.03| 0.03  | 0.17   | 0.03      | 0.00 | 0.03   |
> | ours w/o ENV-VC| 0.67   | 0.43 | 0.33 | 0.27   | 0.13| 0.07  | 0.27   | 0.07      | 0.23 | 0.03   |
> | ours w/o FK-QA | 0.63   | 0.53 | 0.43 | 0.37   | 0.17| 0.03  | 0.37   | 0.13      | 0.10 | 0.00   |
> | ours      | 0.73   | 0.67 | 0.47 | 0.33   | 0.23| 0.07  | 0.43   | 0.10      | 0.17 | 0.07   |
>
>
> The results in  **Table R1, R2, R3** also validate the association between our ENV-VC, FK-QA benchmarks and the improvement of decision-making process.
> When the ENV-VC task is removed, and we directly utilize visual cues without further instruction tuning, there's a notable drop in performance.
> This clearly demonstrates the critical role of ENV-VC task in enhancing the model to intepret and utilize visual cues effectively.
> Similarly, the exclusion of the FK-QA task yields a performance decrease, emphasizing the essential contribution of these two pre-training tasks in enhancing our model's decision-making capabilities.

---

> > ### Author Response · Authors · 2023-11-21
> > **Responses to Reviewer ijGQ**
> >
> > # 2. Baseline Selection Could Be Improved
> > For the "gpt-assistant" baseline, we develop an interactive planning framework using ChatGPT.
> > In this setup, ChatGPT serves as the planner, formulating skill plans in response to prompts that include task descriptions and environmental observations.
> > This approach involves chain-of-thoughts prompting, where we initially provide the planner with a few-shot demonstration with explanations to guide the initial planning step.
> > To address frequent errors observed during testing, we incorporate specific planning rules into the prompts.
> > As the planning process progresses, the planner encounters various scenarios: invalid skill names, redundant skills (already executed), successful skill execution, or failed skill execution.
> > For each of these cases, we carefully craft language feedback and ask the planner to re-plan based on inventory changes.
> >
> > In addition to ``gpt-assistant'', we conduct experiments to compare with another game planning approach for Minecraft: Plan4MC [1].
> > Similar to our Steve-Eye, this approach utilizes a similar strategy to break down a given Minecraft task into several skills, and execute these skills one by one.
> > By employing a specialized skill graph, Plan4MC guarantees the generation of executable plans with 100\% certainty, thereby serving as an upper bound for our approach.
> > As shown in **Table R1, R2, R3** earlier, our Steve-Eye presents competitive performance when compared to Plan4MC,
> > This is noteworthy considering Steve-Eye uses a significantly less optimal LLM (LLaMA-2) compared to ChatGPT used by Plan4MC.
> >
> > In our analysis, we chose not to compare our Steve-Eye with methods like Voyager, as there are significant differences in their underlying mechanisms.
> > Voyager controls the agent by generating code for the Mineflayer API using GPT-4, a process that is inherently different from Steve-Eye, which is driven by a considerably lighter-weight LLM and employs pre-defined RL policies for each skill control.
> > Additionally, Voyager heavily relies on curated prompting for its planning and skill selection process, a feature not as prominently utilized in our approach.
> > Given these fundamental differences, a direct comparison between Steve-Eye and Voyager would not be equitable.
> > However, it's crucial to recognize that Steve-Eye is compatible with Voyager and other similar methods.
> > Voyager uses LLMs to translate an agent's observations into text, while Steve-Eye excels at interpreting multimodal cues.
> > This capability positions Steve-Eye as a potential multimodal foundation model to replace the LLMs in methods like Voyager.
> >
> >
> > [1] Plan4MC: Skill Reinforcement Learning and Planning for Open-World Minecraft Tasks
> >
> > # 3. Clarification on Image Output Utilization Needed
> > Thank you for this suggestion.
> >
> > Firstly, integrating visual generation into our pre-training introduces a novel task that significantly broadens the scale and diversity of our instruction-following dataset.
> > This expansion helps prevent oversimplification of the training process and ensures that the model consistently produces accurate and coherent responses, especially when dealing with complex environments.
> >
> > Secondly, as discussed in our introduction, we argue that multimodal outputs offer a more intuitive way to represent the world, leading to an effective interface for interaction with human users.
> > Therefore, although the multimodal output in this work is not generated for specific tasks,
> > we believe that this is crucial for developing a general-purpose game assistant.
> >
> > # 4. Figure 1 Requires Better Visual Clarity
> > Thank you for your suggestion.
> > We have improved the readability of our figures and address the minor issues in our revised version.

---

> > > ### Author Response · Authors · 2023-11-22
> > > **We sincerely hope that our rebuttal has addressed your concerns. Could you please confirm this?**
> > >
> > > Dear Reviewer ijGQ, thanks for your time in reviewing our work. As the author-reviewer discussion period is expected to end on tomorrow (Nov 22nd), we wonder if you can kindly check whether our rebuttal resolves your concerns (e.g, in-depth analysis on skill planning).
> > >
> > > We are more than happy to have further discussions with the reviewer if there are any remaining issues.

---

> > > > ### Comment · Reviewer_ijGQ · 2023-11-22
> > > > **Post-Rebuttal Comment**
> > > >
> > > > We thank the author for the response, which addresses my concern and I raise the score from 5 to 6.
> > > > Generally, the paper provides a dataset and model that looks valuable to the community. Will the paper also release the model checkpoint?

---

> > > > > ### Author Response · Authors · 2023-11-22
> > > > > **Responds to Reviewer ijGQ**
> > > > >
> > > > > Thank you for your positive feedback!
> > > > > Yes, we are preparing the improved dataset, framework, checkpoints and evaluation benchmark.

---

> > > > > ### Author Response · Authors · 2023-11-23
> > > > > **Post-Respond to Reviewer ijGQ**
> > > > >
> > > > > Thank you again for agreeing to raise the score.
> > > > > We've been checking OpenReview but haven't noticed any updates on the score yet.
> > > > > As the discussion phase is ending soon, could you kindly confirm if the changes have been made?
> > > > > Your assistance is greatly appreciated.

---

### Official Review · Reviewer_m8nW · 2023-11-01

**Soundness:** 2 fair
**Presentation:** 2 fair
**Contribution:** 3 good
**Rating:** 6
**Confidence:** 4

**Summary:**

The proposed method seeks to overcome "text-only" problems by combining an LLM with a visual encoder, which processes both visual and textual inputs and produces multimodal responses. This model is trained on a new dataset proposed by the authors, which contains multimodal perception, a foundational knowledge base, and skill prediction & planning. The trained model can perform multimodal I/O, benefitting from the two-stage training and fine-tuning.

The effectiveness of Steve-Eye is demonstrated through three proposed new benchmarks: environmental visual captioning (ENV-VC), foundational knowledge question answering (FK-QA), and skill prediction and planning (SPP).  The results show that finetuned Steve-Eye surpasses existing LLM in Minecraft scenarios, especially in multimodal Q&A.

**Strengths:**

Included in the summary and questions section.

**Weaknesses:**

Included in the questions section.

**Questions:**

I'm still confused about the Multimodal Perception Instructions section. Are the responses for the instructions drawn from those processed dense captions? How exactly is the extraction done? Also, it would be clearer for readers if example instructions, responses, and snapshots were provided together, as the current format requires flipping back and forth between the text and the appendix to understand these benchmarks.

For foundational knowledge question answering, evaluation relies on ChatGPT, which may be hard to reproduce and could incur high costs. Moreover, I doubt about the evaluation performance. Given that Steve-Eye-13b is fine-tuned on a vast amount of domain data, it should theoretically perform much better than Llama. Could the limited performance boost be due to the evaluation method or insufficient training on Wikipedia data?

Regarding skill prediction, it seems that rule-based analysis is necessary since simply combining two random frames likely won't give enough information for skill assessment, even with human expertise. Yet, this kind of pretraining appears to somewhat improve performance, according to results from the authors. However, I believe a different approach, like training the model to predict the next move from videos and instructions, might be more beneficial, which requires a stronger capability to process sequences of frames or videos.

In summary, the proposed Steve model shows promise in multimodal Q&A for Minecraft, but it's not yet a fully embodied agent. Future work could improve its in-context Q&A and control capabilities, which would allow for understanding temporal context and environmental feedback-based control.

Though there's potential to refine the evaluations and finetuning tasks, the authors do contribute massive datasets and pretrained multimodal large models to the community, which they plan to make publicly available. Hence, I am giving a positive review.

---

> ### Author Response · Authors · 2023-11-21
> **Responds to Reviewer m8nW**
>
> # 1. Details of Multimodal Perception Instructions
>
> Thank you for your suggestions!
> Our generated responses to instructions can be divided into two segments:
> * (1) Responses directly derived from processed dense captions.
> To foster data diversity, we instruct ChatGPT to prompt a variety of description templates as shown in Appendix 1.1.
> Each caption is then rephrased by randomly selecting one of these templates, ensuring a broad spectrum of expressions.
> * (2) Responses to questions about the caption.
> We use ChatGPT to curate a set of question formats and collect QA pairs accordingly.
> The model is required to accurately respond based on the selected question and the visual content of the caption.
>
> In addition, due to the space limit, we cannot provide all examples, responses, and snapshots together in the main text. We will make the paper more readable in the next version.
>
> # 2. Questions about Foundational Knowledge Question Answering
> Fristly, to ensure reproducibility, we will release our 1,000 QA evaluation benchmark.
> This evaluation process is cost-efficient, as the usage of gpt-turbo-3.5 charges only $0.0015$ per 1,000 tokens.
> Consequently, the entire evaluation costs less than $1$ dollar.
>
> Secondly, it's important to acknowledge the existence of noises within our Wiki data, particularly those extracted from Wiki Tables.
> As you can see in https://minecraft.fandom.com/wiki, Wiki data is highly structured, which inevitably causes data noises even with careful processing.
> We attribute the limited performance boost to such data noises.
> To address this issue, we are dedicating more resources to refine the collected Wiki data for a better-quality version.
> Given that such an endeavor has not been previously undertaken, we believe that offering this knowledge-centric resource will significantly benefit the community.
>
> # 3. Questions about Skill Prediction and Future Work
> Insightful suggestion!
>
> Firstly, regarding the use of rule-based analysis, it's not directly employed in our skill prediction evaluation but plays an important role during the pre-training of our ENV-VC task.
> The ENV-VC task is tailored to enable our model to interpret the current status primarily via visual cues.
> We acknowledge that our skill prediction lacks explicit rule-based information.
> However, we believe relying solely on visual cues for predictions is more in line with human's decision-making process compared to a dependency on rule-based analysis, since there won't be explicit rules in our real life.
>
> Secondly, our initial attempt to predict moves from videos and instructions has proven to be a substantial challenge and did not yield satisfactory results before.
> As a result, currently our Steve-Eye assesses only the success or failure of skill execution at each step, in order that Steve-Eye can decide whether to re-plan.
> It does not engage in predicting low-level moves.
> Instead, we implement low-level moves by additionally training an RL-based policy for each involved skill.
> Nevertheless, we agree that directly predicting low-level moves based on visual cues could be more advantageous for creating realistic embodied agents.
> Thus, we are committed to exploring this direction in our future work.

---

> > ### Author Response · Authors · 2023-11-22
> > **We sincerely hope that our rebuttal has addressed your concerns. Could you please confirm this?**
> >
> > Dear Reviewer m8nW, thanks for your time in reviewing our work.
> > As the author-reviewer discussion period is expected to end on tomorrow (Nov 22nd), we wonder if you can kindly check whether our rebuttal resolves your concerns.
> >
> > We are more than happy to have further discussions with the reviewer if there are any remaining issues.

---

> ### Comment · Reviewer_m8nW · 2023-11-22
> **Reply to the authors' rebuttal**
>
> It's good to know that the usage of gpt-turbo-3.5 charges only per 1,000 tokens. However, since gpt-turbo-3.5 is a commercial API, it's not sure whether it will always be available for the academic community to reproduce the evaluation. I still think an alternative evaluation which can be done offline or independent from third-party service is better for reproduction and follow-up works.
>
> Thanks to the authors for the reply regarding the questions I raised. Overall it somehow solves my other doubts about the manuscript. I have no further questions at this point.

---

> > ### Author Response · Authors · 2023-11-23
> > **Reply to reviewer m8nW**
> >
> > Thanks for your reply.
> > Indeed, relying solely on the GPT API for evaluation carries certain risks. To mitigate this, we propose developing an alternative QA evaluation set designed for direct assessment using generated ground truths, such as Yes/No or multiple-choice options (A/B/C/D, etc.). I believe this approach will effectively address your concerns.

---

### Meta-Review · Area_Chair_G9ss · 2023-12-07

**Metareview:**

"Steve-Eye" presents an approach to integrating multimodal perception into embodied agents for game playing, particularly in the Minecraft environment. The reviewers identify that the integration of a visual encoder with an LLM, allowing for multimodal input and output, is acknowledged as a significant advancement in the field of embodied agents. The model is evaluated on newly proposed benchmarks (ENV-VC, FK-QA, and SPP), Steve-Eye demonstrates superiority in multimodal question answering and planning in Minecraft scenarios. Meanwhile, the reviewers raised concerns, especially in baseline selection and comparative evaluation.

The paper revceived borderline ratings, with three borderline accepts and one borderline reject (One reviewer has promised to upgrade the score to borderline accept, but seems to forget to update. The AC counts it as borderline accept here.).

The AC checked all the materials. The AC believes the authors have well addressed the concerns, and the strengths of the paper outweight its shortcomings. Thus, the paper is accepted.

**Justification For Why Not Higher Score:**

Reviewers noted a discernible lack of detailed discussion and analysis in skill planning, especially compared to other game-playing agents. Some reviewers raised concerns about the selection of baselines for performance evaluation.

**Justification For Why Not Lower Score:**

The reviewers identify that the integration of a visual encoder with an LLM, allowing for multimodal input and output, is acknowledged as a significant advancement in the field of embodied agents. The model is evaluated on newly proposed benchmarks (ENV-VC, FK-QA, and SPP), Steve-Eye demonstrates superiority in multimodal question answering and planning in Minecraft scenarios.

---

### Decision · Program_Chairs · 2024-01-16

Accept (poster)